# Mesenchymal stromal cell apoptosis is required for their therapeutic function

Swee Heng Milon Pang [1], Joshua D'Rozario [1,2,7], Senora Mendonca[1,7], Tejasvini Bhuvan[1], Natalie L. Payne [3], Di Zheng[1], Assifa Hisana[1], Georgia Wallis[1], Adele Barugahare [4], David Powell [4], Jai Rautela [2], Nicholas D. Huntington [2], Grant Dewson[5,6], David C. S. Huang [5,6], Daniel H. D. Gray [5,6] & Tracy S. P. Heng [1✉]

Multipotent mesenchymal stromal cells (MSCs) ameliorate a wide range of diseases in preclinical models, but the lack of clarity around their mechanisms of action has impeded their clinical utility. The therapeutic effects of MSCs are often attributed to bioactive molecules secreted by viable MSCs. However, we found that MSCs underwent apoptosis in the lung after intravenous administration, even in the absence of host cytotoxic or alloreactive cells. Deletion of the apoptotic effectors BAK and BAX prevented MSC death and attenuated their immunosuppressive effects in disease models used to define MSC potency. Mechanistically, apoptosis of MSCs and their efferocytosis induced changes in metabolic and inflammatory pathways in alveolar macrophages to effect immunosuppression and reduce disease severity. Our data reveal a mode of action whereby the host response to dying MSCs is key to their therapeutic effects; findings that have broad implications for the effective translation of cell-based therapies.

[1] Department of Anatomy and Developmental Biology, Biomedicine Discovery Institute, Monash University, Clayton, VIC 3800, Australia. [2] Department of Biochemistry and Molecular Biology, Biomedicine Discovery Institute, Monash University, Clayton, VIC 3800, Australia. [3] Australian Regenerative Medicine Institute, Monash University, Clayton, VIC 3800, Australia. [4] Monash Bioinformatics Platform, Monash University, Clayton, VIC 3800, Australia. [5] The Walter and Eliza Hall Institute of Medical Research, Parkville, VIC 3052, Australia. [6] Department of Medical Biology, University of Melbourne, Parkville, VIC 3010, Australia. [7]These authors contributed equally: Joshua D'Rozario, Senora Mendonca. ✉email: Tracy.Heng@monash.edu

Multipotent mesenchymal stromal cells (MSCs) are a heterogeneous population of cells isolated from bone marrow and other tissue stroma that have immuno-suppressive and anti-inflammatory properties. In many animal models of disease, MSCs have demonstrated therapeutic efficacy regardless of major histocompatibility complex or species barriers[1]. MSCs are approved for the treatment of acute graft-versus-host disease (GvHD) and Crohn's fistula in some countries, and are being investigated in many other disease conditions[2,3]. Recent efforts have focused on using MSCs to treat acute respiratory distress syndrome (ARDS) in COVID-19 patients, due to the beneficial effects of MSC administration on lung injury and inflammation[4]. It remains unclear how MSCs isolated from different tissues or species could exert therapeutic effects on such a wide range of unrelated diseases.

The current consensus is that therapeutic applications of MSCs are based on their secretion of a wide array of cytokines, chemokines and subcellular particles[5]. However, MSCs do not engraft[6] and there is little evidence that these cells even survive infusion or injection[7]. Studies tracking MSCs after intravenous (i.v.) administration reported lung entrapment, upregulation of apoptosis-associated genes and presence of apoptotic bodies in the lungs[7–9]. Only dead MSCs were detected in the lungs and liver 24 h after administration[10]. In a GvHD study, it was demonstrated that only patients whose immune cells were able to induce apoptosis in MSCs responded to MSC therapy, suggesting that MSC apoptosis may contribute to clinical response[11].

Apoptosis is an 'immunologically silent' form of cell death, involving proteolysis of hundreds of cellular proteins by caspases that can be activated via two distinct albeit converging pathways[12]. The intrinsic (mitochondrial) pathway is triggered by cellular stressors (e.g. cytokine deprivation or chemotherapeutic drugs) that alter the balance between pro-apoptotic BH3-only proteins and anti-apoptotic members of the BCL-2 family towards the former. Consequently, the effectors of apoptosis, BAK and BAX, are activated, permeabilising the mitochondrial outer membrane to release cytochrome $c$ to activate caspase-9. This, in turn, triggers the 'executioner' caspases 3, 6 and 7, which demolish the cell without releasing danger signals[12]. The alternative, extrinsic (death-receptor) pathway is initiated by ligation of members of the Tumor Necrosis Factor (TNF) receptor family that contain an intracellular 'death domain' (e.g. FAS, TNFR1). Pro-death ligands (e.g. FASL, TNF) trimerise these receptors, leading to the activation of caspase-8, which in turn activates the 'executioner' caspases[12]. In contrast to apoptosis, necroptosis and pyroptosis are two forms of lytic cell death that cause inflammation and have evolved to combat certain pathogen escape mechanisms by activating IL-1 and IFN responses[13].

Apoptotic lymphocytes have been shown to dampen cell-mediated and humoral immune responses[14,15]. Similarly, apoptotic MSCs induced immunosuppressive effects in animal models of lung injury and inflammation[16–18]. Macrophages and monocytes that had ingested apoptotic cells produced anti-inflammatory mediators, such as TGF-β, IL-10, indoleamine 2,3-dioxygenase (IDO) and prostaglandin E2 (PGE$_2$)[10,19–22]. Importantly, depletion of macrophages has been shown to abrogate the beneficial effects of MSC therapy in multiple disease models[11,23,24]. In light of these studies, an open question in the field is whether the broad immunosuppressive effects of i.v. administered MSCs are solely mediated by the host response to the apoptosis of MSCs, rather than an intrinsic property of viable MSCs.

In the current study, we generated apoptosis-refractory human MSCs to test whether inhibiting cell death in MSCs would abrogate their therapeutic efficacy, thereby establishing that apoptosis of MSCs is necessary for the immunomodulatory effects exerted by their infusion. Our data demonstrated that MSC apoptosis and subsequent efferocytosis are required for their full immunosuppressive effects in vivo, answering the long-standing question of how MSCs mediate therapeutic effects that persist beyond their survival.

## Results

**i.v. administered MSCs rapidly undergo apoptosis in the lungs of recipient mice.** Pulmonary conditions benefit from MSCs delivered via the i.v. the route, due to lung entrapment. We previously demonstrated that human MSCs inhibited allergic asthma despite their lack of sustained persistence in the lungs[24]. It is possible that our method of detecting MSCs (via bioluminescence imaging of luciferase-expressing MSCs) was not sensitive enough to detect very small numbers of MSCs which could have potentiated the inhibitory effects in the lungs. Therefore, we undertook a detailed time-course analysis of labelled MSCs by flow cytometry to determine the fate of these cells. Human bone marrow (BM)-derived MSCs were labelled with CellTrace™ Violet (CTV) and injected into BALB/c recipient mice (Fig. 1a). At various timepoints postinjection, the lungs of recipient mice were digested and analysed via flow cytometry to track the CTV-labelled MSCs. Using this method, we could detect $1 \times 10^6$ CTV-labelled MSCs in the lung confidently, down to $1 \times 10^5$ injected MSCs (Supplementary Fig. 1a). CTV label was detected in both CD45$^-$ stromal and CD45$^+$ haematopoietic populations in the lungs (Fig. 1b). CD73$^+$ MSCs were contained within the CD45$^-$CTV$^{hi}$ population, which exhibited comparable fluorescence signal to the input (Fig. 1a, b). This CD45$^-$CTV$^{hi}$ population showed high levels of caspase 3 activation within 1 h postinjection, indicating apoptosis of the MSCs (Fig. 1c). The CTV$^{lo}$CD45$^-$ population also displayed caspase 3 activation at 1 h postinjection, although not to the same extent as the CTV$^{hi}$CD45$^-$ population (Fig. 1b). This CTV$^{lo}$CD45$^-$ population contained lung epithelial cells that had engulfed apoptotic CTV-labelled MSCs (Supplementary Fig. 1b), as previously shown[25]. The proportion of CTV$^+$ MSCs within the CD45$^-$ population decreased progressively over time, such that only a very small amount was detected at 8 h (Fig. 1d).

MSCs labelled with a different fluorescent tracking dye, CellTracker™ Orange CMTMR, similarly underwent apoptosis and decreased in number progressively with time, confirming our observations (Fig. 1e). Rapid caspase 3 activation was also seen in MSCs derived from human umbilical cord tissue and adipose tissue, indicating that apoptosis in the lung was common to MSCs from various tissue sources (Fig. 1f). These data indicate that i.v. injected MSCs that lodge in the lungs promptly undergo apoptosis and are cleared within 24 h.

**Apoptotic MSCs retain immunosuppressive capacity in vivo.** To investigate whether there is a requirement for viable MSCs to persist in the lungs for their immunomodulatory effects, we compared the capacity of viable versus apoptotic MSCs to inhibit OVA-induced asthma. OVA-sensitised mice were injected with MSCs that had been pre-treated with 0.5 μM staurosporine (STS-MSCs) for 6 h to induce apoptosis (Fig. 2a). At this dose of staurosporine, MSCs were mostly viable at the time of injection, but became AnnexinV$^+$PI$^+$ after an additional 12 h in staurosporine-free culture medium. Upon airway challenge with OVA, administration of STS-MSCs inhibited eosinophil influx in BALF and OVA-specific IL-5 and IL-13 production to a similar extent as untreated MSCs (Fig. 2b), indicating that the inhibitory effects of MSCs in the lungs do not require MSCs to remain viable.

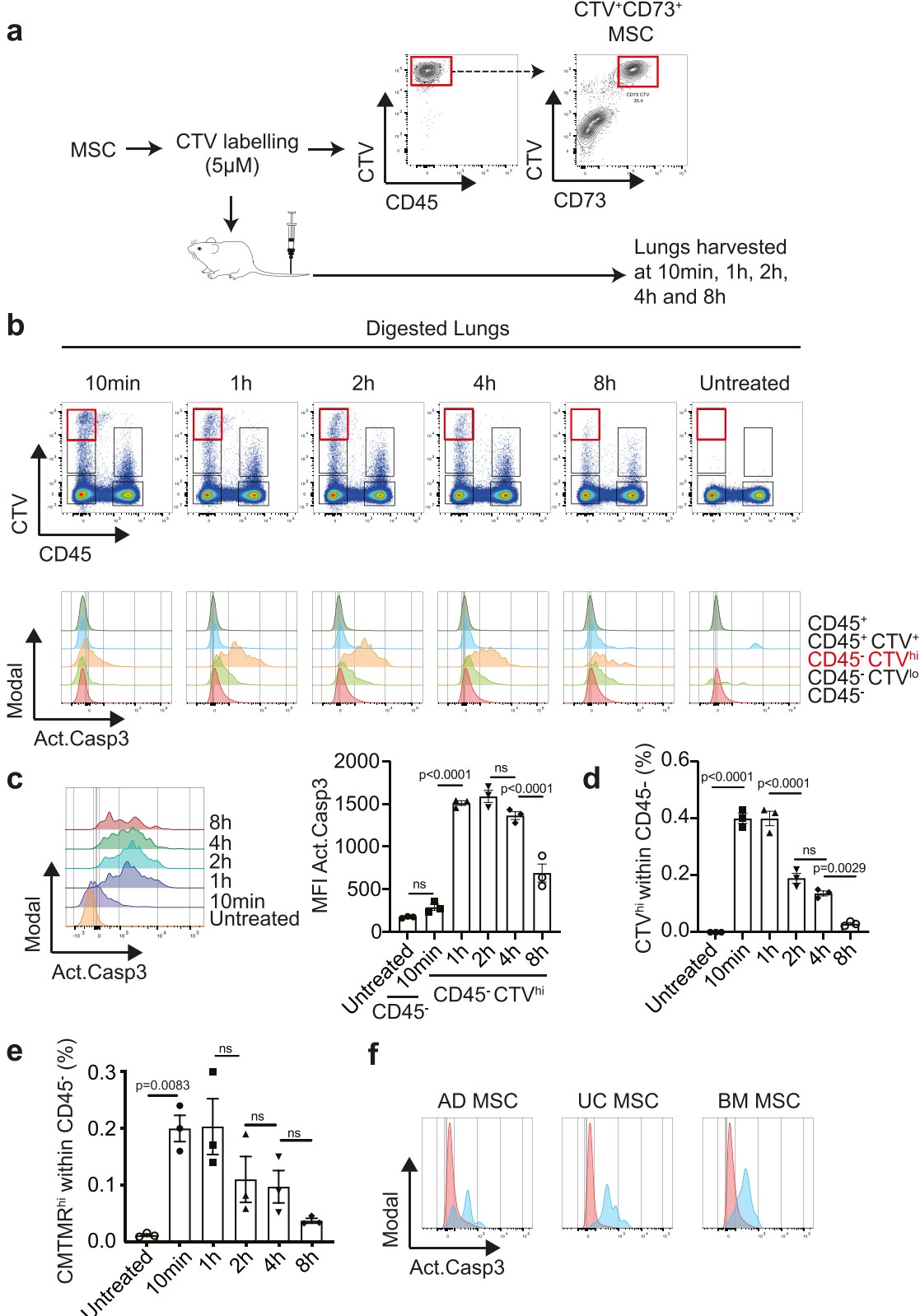

In a similar experiment, MSCs were treated with BH3-mimetic drugs that inhibit pro-survival BCL-2 proteins to selectively induce apoptosis via the intrinsic pathway. Human MSCs underwent apoptosis following treatment with a combination of BH3-mimetic drugs targeting BCL-2, BCL-XL and MCL-1 (BH3-MSCs) (Fig. 2c). Drugs targeting these molecules singly were ineffective, indicating different pro-survival BCL-2 proteins

support MSC survival (Fig. 2c). OVA-sensitised mice were then injected with live (DMSO-treated) or apoptotic BH3-MSCs prior to airway challenge with OVA (Fig. 2d). Both MSC treatment groups showed a decrease in lung eosinophils and OVA-specific IL-5 and IL-13 production compared to OVA-sensitised mice that did not receive cell treatment (Fig. 2e). Airway hyperresponsiveness (AHR), indicated by increased lung resistance (Rl) and

**Fig. 1 MSCs undergo apoptosis in the lungs shortly after i.v. administration. a** MSCs were labelled with CTV and the fluorescence intensity was measured prior to injection to guide gating of MSCs re-isolated from mouse lungs after injection. CTV$^+$ MSCs were CD45$^-$CD73$^+$. **b** Top panel shows representative flow cytometric plots of different populations identified by CTV and CD45 markers in lung samples at various timepoints after MSC injection. CD45$^-$CTV$^{hi}$ MSCs are indicated by the red box, as shown in **a**. Lower panel shows expression levels of activated caspase 3 in CD45$^-$CTV$^-$, CD45$^-$CTV$^{lo}$, CD45$^-$CTV$^{hi}$, CD45$^+$CTV$^-$ and CD45$^+$CTV$^+$ populations. **c** Expression levels and mean fluorescence intensity (MFI) of activated caspase-3 on CD45$^-$ (from untreated mice that did not receive MSCs) and CD45$^-$CTV$^{hi}$ (from mice that received CTV$^+$ MSCs) at various timepoints, as shown in **b**. Data expressed as mean ± SEM, $n = 3$ mice per group over three independent experiments. **d** Frequency of CTV$^{hi}$ within the CD45$^-$ population as shown as in **b**. Data expressed as mean ± SEM, $n = 3$ mice per group over three independent experiments. **e** Frequency of CMTMR$^{hi}$ within the CD45$^-$ population in lungs from mice that received CMTMR-labelled MSCs in a separate experiment. Data expressed as mean ± SEM. $n = 3$ mice per group. **f** Expression levels of activated caspase 3 in CTV-labelled adipose (AD), umbilical cord (UC) or bone marrow (BM) MSCs re-isolated from mouse lungs 4 h after i.v. injection. Data representative of two independent experiments. $p$ values by one-way ANOVA (Tukey's post hoc test); ns not significant. Source data are provided as a Source Data file.

decreased dynamic compliance (Cdyn) in response to increasing doses of methacholine, was also inhibited in mice that received live or BH3-MSCs (Fig. 2f). Lung tissue sections from mice that received live or BH3-MSCs showed a reduction in inflammatory cells and mucus-secreting goblet cells in airway tissue (Fig. 2g).

Conversely, mouse embryonic fibroblasts (MEFs), which were relatively more resistant to apoptosis induction and required higher concentrations and longer duration of BH3-mimetic drug treatment (Supplementary Fig. 2a), failed to inhibit the hallmark features of allergic asthma (Supplementary Fig. 2b–e).

Collectively, these data show that apoptotic MSCs exerted immunosuppressive effects in the lungs and inhibited allergic asthma to a similar extent as administration with viable MSCs, while MEFs which are relatively resistant to apoptosis, did not inhibit asthma.

**Combined BAK/BAX deficiency renders MSCs resistant to apoptosis and attenuates their immunosuppressive capacity in vivo.** As caspase 3 activation in MSCs was evident as early as 1 h post-i.v. injection (Fig. 1c), and administration of apoptotic MSCs induced immunosuppressive effects in the lungs (Fig. 2e, f), we investigated the impact of inhibiting apoptosis in MSCs on their therapeutic efficacy. In the intrinsic pathway of apoptosis, activation of BAK and/or BAX generates pores in the outer mitochondrial membrane; an event considered the "point of no return" because it triggers an amplified caspase activation cascade[12]. Therefore, to generate apoptosis-resistant MSCs, we genetically deleted *BAK/BAX* utilising Cas9 ribonucleoprotein (RNP) complexes introduced via electroporation. BAK/BAX-targeted MSCs (BKX-MSCs) retained the typical characteristics of MSCs, including surface phenotype, capacity for differentiation into mesenchymal lineages in vitro, ability to form fibroblastic CFUs and the capacity to inhibit T cell proliferation in vitro (Supplementary Fig. 3a–d).

Treatment of BKX-MSCs with BH3-mimetic drugs showed that > 30% of BKX-MSCs were resistant to apoptosis induction (i.e. AnnexinV$^-$) compared to less than 2% of control MSCs (Fig. 3a). Culture expansion of surviving BKX-MSCs and subsequent BH3-mimetic drug treatment selected for MSCs with increased resistance to apoptosis induction (> 70% AnnexinV$^-$). Treatment with a higher concentration of BH3-mimetic drugs further selected for apoptosis-resistant cells, such that 98% of BKX-MSCs remained viable (AnnexinV$^-$) (Fig. 3a). Selection for apoptosis-resistant MSCs correlated with a reduction in BAX and BAK proteins (Fig. 3b). Compared to nontargeted MSCs which underwent caspase 3 activation within 1 h postinjection, the majority of BKX-MSCs did not show caspase 3 activation in the lungs, indicating that the predominant path to death of MSCs was mitochondrial apoptosis (Fig. 3c). Furthermore, treatment of MSCs with mouse serum did not result in increased cell death at

10 mins or 1 h (Fig. S3e). Both control MSCs and BKX-MSCs exhibited the same amount of cell death after 24 h of culture with mouse serum, an effect that was inhibited by heat-inactivation of serum complement (Fig. S3e). Our data indicate that complement was not the main driver of MSC apoptosis that occurs in the lung within 1 h post injection.

Next, to test whether apoptosis-resistant MSCs retained immunosuppressive capacity in vivo, OVA-sensitised mice received i.v. injections of BKX-MSCs or control MSCs prior to airway challenge with OVA. T cell responsiveness to OVA restimulation, measured by OVA-specific T cell proliferation in 2 different ways (CFSE dilution and MTS bioreduction), showed a decrease in proliferative response in OVA-sensitised mice treated with control MSCs but not BKX-MSCs (Fig. 3d). The reduction in lung eosinophils and increase in AMs, previously reported in MSC-treated mice[24], was less evident but still significant in BKX-MSC-treated mice (Fig. 3e). Control MSCs, but not BKX-MSCs, reduced OVA-specific IL-5 and IL-13 (Fig. 3e). Importantly, combined BAK/BAX deficiency in MSCs was sufficient to reduce their therapeutic efficacy on the lung function of OVA-sensitised mice, as administration of control MSCs, but not BKX-MSCs, inhibited AHR as measured by Rl and Cdyn (Fig. 3f). The diminished effectiveness of BKX-MSCs was also evident histologically, as lung tissue sections showed a reduction in inflammatory cells and mucus-secreting goblet cells in the airways of MSC-treated mice, but to a lesser extent in BKX-MSC-treated mice (Fig. 3g).

We also examined whether the reduction in immunosuppressive capacity in BKX-MSCs could be observed in a disease model where tissue inflammation occurs at a site distal to the lung. Myelin oligodendrocyte glycoprotein (MOG)-induced experimental autoimmune encephalitis (EAE) is a mouse model of multiple sclerosis shown to benefit from MSC therapy[26]. EAE mice that received control MSCs during the induction phase of disease, when priming of MOG-specific CD4$^+$ T cells occurs, showed a reduction in the daily mean clinical score compared to vehicle (PBS)-treated EAE mice (Fig. 3h), as previously demonstrated[27]. EAE mice that received BKX-MSCs, on the other hand, exhibited a delay in onset of clinical signs but developed disease that was as severe as untreated EAE mice (Fig. 3h). These results were also reflected upon histological analysis of spinal cord tissue at D29 after disease induction, which showed reduced leucocyte infiltration and demyelination within the spinal cord parenchyma only in mice that received control MSCs (Fig. 3i). Notably, the ability of control MSCs and BKX-MSCs to reduce MOG-specific T cell proliferative response was comparable at D9 after disease induction, but by D29 this inhibitory effect was only maintained in mice that had received control MSCs (Fig. 3j). This may be reflective of the importance in regulating both the adaptive and innate immune responses

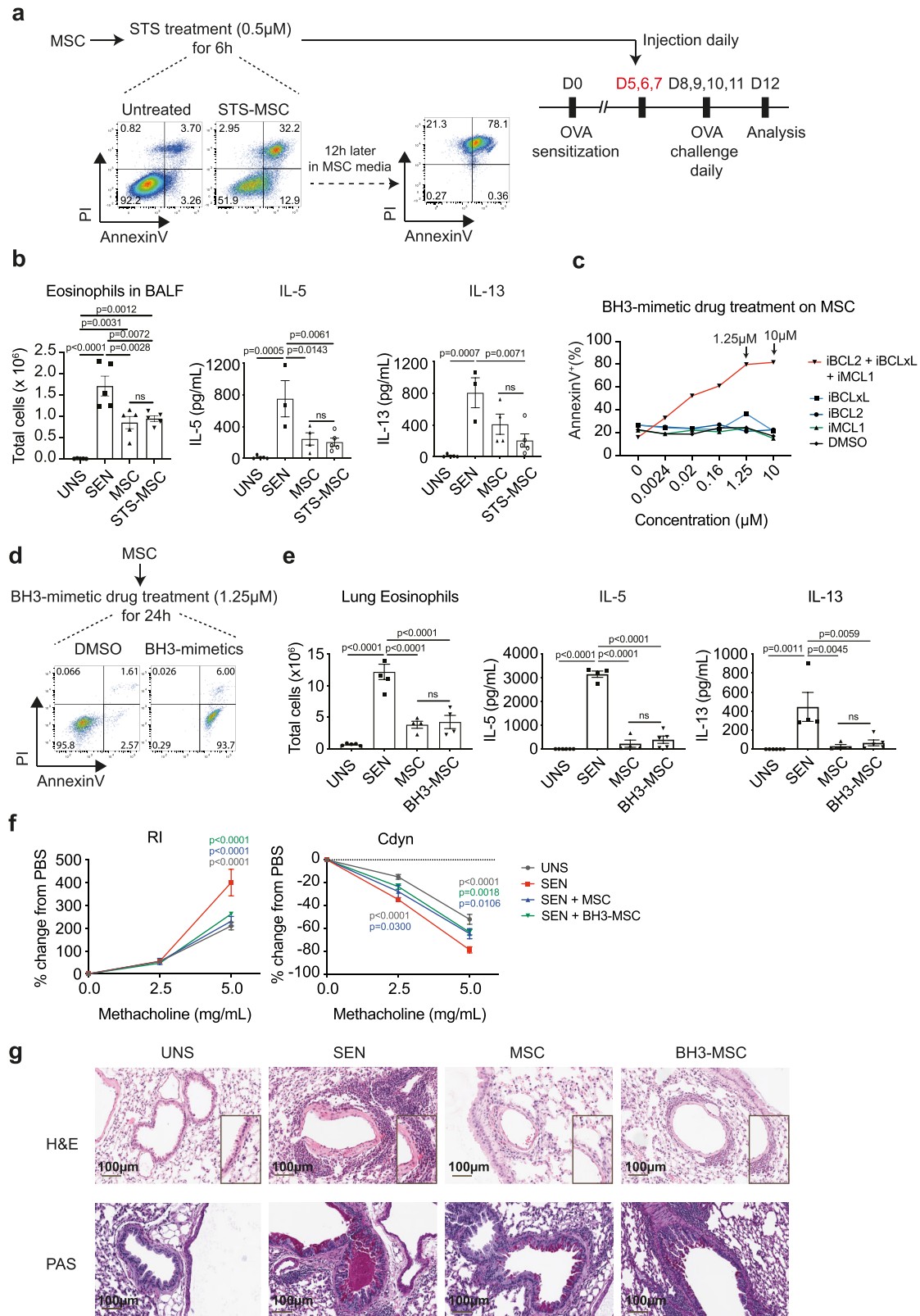

during EAE. Importantly, control MSCs, but not BKX-MSCs, reduced circulating inflammatory Ly6C[hi] monocytes that are required to cause tissue damage following their infiltration into the CNS[28], while neutrophils numbers were unchanged (Fig. 3k). Thus, the data showed that the inability of BKX-MSCs to reduce circulating inflammatory monocytes that infiltrate the CNS to cause tissue damage, led to subsequent failure to curb disease.

Together, these findings indicate that apoptosis of MSCs is necessary for efficient in vivo immunosuppression and therapeutic efficacy.

**MSC clearance from the lung is not dependent on cytotoxic cells or allorecognition.** Despite demonstrating therapeutic efficacy in various animal models of disease, most studies have

**Fig. 2 Apoptotic MSCs retain their immunosuppressive capacity in vivo. a** MSCs were treated with 0.5 µM of STS for 6 h before i.v. administration into OVA-sensitised mice prior to OVA challenge. Flow cytometric plots of AnnexinV vs. PI staining in MSCs after treatment with STS for 6 h, and at 12 h later following removal from STS. **b** Number of eosinophils in BALF (Data expressed as mean ± SEM; n = 5 mice per group), and IL-5 and IL-13 production by DLN cells in response to OVA restimulation. UNS = unsensitised mice; SEN = OVA-sensitised mice; MSC = OVA-sensitised mice that received MSCs; STS-MSC = OVA-sensitised mice that received STS-treated MSCs. Data expressed as mean ± SEM (UNS n = 5; SEN n = 3; MSC n = 4; STS-MSC n = 5). p values by one-way ANOVA (Tukey's post hoc test); ns, not significant. **c** Frequency of AnnexinV+ MSCs 72 h following treatment with different concentrations of individual BH3-mimetic drugs inhibiting MCL-1, BCL-2 or BCL-XL, or a combination of all three. Data representative of two independent experiments. **d** MSCs were treated with 1.25 µM of BH3-mimetic drugs for 24 h to induce apoptosis before i.v. administration into OVA-sensitised mice. **e** Total number of eosinophils in lungs (Data expressed as mean ± SEM; UNS n = 5; SEN n = 5; MSC n = 4; BH3-MSC n = 4), and DLN OVA-specific IL-5 and IL-13 production. (Data expressed as mean ± SEM; UNS n = 6; SEN n = 4; MSC n = 4; BH3-MSC n = 5). p values by one-way ANOVA (Tukey's post hoc test); ns, not significant. **f** RI (UNS n = 3; SEN n = 4; MSC n = 5; BH3-MSC n = 5) and Cdyn (UNS n = 5; SEN n = 5; MSC n = 4; BH3-MSC n = 5) in response to increasing doses of methacholine on Day 12. Data expressed as mean ± SEM; p values by two-way ANOVA (Tukey's post hoc test), compared with SEN. **g** Lung sections were stained with H&E and PAS for pulmonary inflammation and mucus production respectively. Magnification 20x, scale bar = 100 µm. Histological images were representative of five mice per group. Source data are provided as a Source Data file.

difficulty detecting significant numbers of human MSCs in the lung or in other tissues shortly after i.v. administration[1]. We sought to determine whether immune mechanisms were required for the clearance of infused MSCs. Whole-body and direct imaging of lungs dissected from mice injected with MSCs expressing firefly luciferase (fluc) revealed the presence of luciferase signals up to, but not beyond, 3 days post-i.v. injection, confirming the short lifespan of i.v. administered MSCs (Fig. 4a). In a separate experiment tracking, MSCs in OVA-sensitised mice challenged with airway exposure to OVA showed that MSCs were cleared at a similar rate to MSCs in unsensitised mice (Fig. 4b). This finding indicates that the presence of lung inflammation does not extend the survival of MSCs in the lungs.

We next investigated whether MSC apoptosis and clearance is due to an immune response against xenogeneic or allogeneic cells. Human and mouse BM-derived MSCs were transduced with a bicistronic lentiviral vector encoding fluc and eGFP[24] to enable FACSorting of fluc-GFP+ MSCs for bioluminescence imaging of BALB/c mice following injection of the same number of MSCs. Human MSCs (xenogeneic), BALB/c MSCs (syngeneic) or C57BL/6 MSCs (allogeneic) administered into BALB/c mice were cleared within a similar timeframe of 1–2 days as detected by whole-body bioluminescence imaging of live animals (Fig. 4c). On average, human MSCs were cleared a day earlier than mouse MSCs in immunocompetent mice, but the absence of persistent syngeneic MSCs suggests that allorecognition does not play a major role in the apoptosis and clearance of i.v. administered MSCs in the lungs.

In a recent study, the immunosuppressive effects of human MSCs in a mouse model of graft-versus-host disease were found to be dependent on cytotoxic cells (likely T cells and NK cells) in recipient mice that induced apoptosis of human MSCs[11]. It is unclear whether this mechanism of MSC apoptosis applies to other models of disease that do not involve cytotoxic cells as disease effector cells, but in which MSCs have demonstrated therapeutic efficacy. Therefore, we tracked the persistence of luciferase-expressing human MSCs in immunocompetent BALB/c mice compared to two immunodeficient mouse models, NOD/SCID/Il2rγc[−/−] (NSG)[29] and BALB/c NOD.sirpa Rag2[−/−] Il2rγc[−/−] (BRGS)[30]. Human MSCs were cleared in all mouse strains within 2–3 days as detected by bioluminescence imaging of live animals (Fig. 4d). Luciferase signals emitted by human MSCs in BALB/c mice were undetectable 2 days post-i.v. injection, while luciferase signals in NSG and BRGS mice were very low 2 days post-i.v. injection and completely undetectable by day 3 (Fig. 4d).

Analysis of immune cell types in the lungs confirmed that T cells, B cells and NK cells were deficient in NSG and BRGS mice but myeloid-derived cells were present in the lungs of both models (Fig. 4e). Specifically, NSG and BRGS mice had increased neutrophils and Ly6C[−] monocytes, but were deficient in Ly6C[+] monocytes. BRGS mice had increased cDC1 cells and an equivalent number of cDC2 cells compared to BALB/c controls, while NSG mice were deficient in both DC subsets. NSG and BRGS mice have the equivalent number of eosinophils and interstitial macrophages (IM) as BALB/c mice, while alveolar macrophages (AM) are decreased but still present in significant numbers in both NSG and BRGS mice (Fig. 4e). Thus, the rapid clearance of MSCs in the absence of an adaptive immune response and cytotoxic cells suggests that MSCs are dying in the lung due to factors unrelated to immune cell function (e.g. this death is more likely driven by nutrient/growth factor deprivation) and getting cleared by myeloid cells, regardless of the inflammatory state.

**MSCs are cleared by phagocytic cells in the lungs.** When viable CTV-labelled MSCs were i.v. injected into BALB/c mice, CTV label was detected in both CD45[−] stromal and CD45[+] haematopoietic populations in the lungs (Fig. 1b). The CTV[+]CD45[+] population contained professional phagocytes that had taken up CTV-labelled MSCs. To determine which lung phagocytes had taken up MSCs, we phenotyped these events with an extensive panel of myeloid cell markers[31] (Fig. 5a). Tracking CTV[+]CD45[+] cells over time revealed that a hierarchy of phagocytic cell types engulfed CTV-labelled MSCs in the lungs (Fig. 5b). At 10 min post-i.v. injection, Ly6G[+] neutrophils were the predominant cell type phagocytosing MSCs, composing 40% of the CTV[+]CD45[+] population. By 1 h, there was more uptake by Ly6C[hi] and Ly6C[lo] monocytes, followed by CD11b[−]CD103[+] cDC1 at 2 h, and CD64[+] IM at 4 h. By 8 h post-i.v. injection, the main phagocytic cells were IM and cDC1. All through the initial 8 h, uptake of CTV-labelled MSCs by CD11c[+]SiglecF[+] AM remained consistent at 5–10% of the CTV[+]CD45[+] population, while uptake by CD11b[+] cDC2 remained consistently low at 1–3%.

Strikingly, CTV[+] neutrophils, monocytes and AMs upregulated MHC class II (Fig. 5C), indicating activation of these cell types upon uptake of apoptotic MSCs. MHC class II upregulation was also observed in AMs from OVA-sensitised mice that were treated with MSCs (Fig. 5C). We therefore sought to confirm efferocytosis of MSCs in these cell types. Viable MSCs were labelled with pHrodo[TM] RED, a pH-sensitive dye which fluoresces bright red in phagosomes due to the acidic pH[32], and injected into BALB/c mice. At 1 h post-i.v. injection of viable MSCs, pHrodo signals were observed in neutrophils and monocytes (both Ly6C[+] and Ly6C[lo/−]), confirming uptake of MSCs by these cells (Fig. 5D). Live-cell imaging further confirmed that pHrodo[TM] RED-labelled apoptotic MSCs, but not BKX-MSCs, were efferocytosed efficiently by AMs (Fig. 5E,

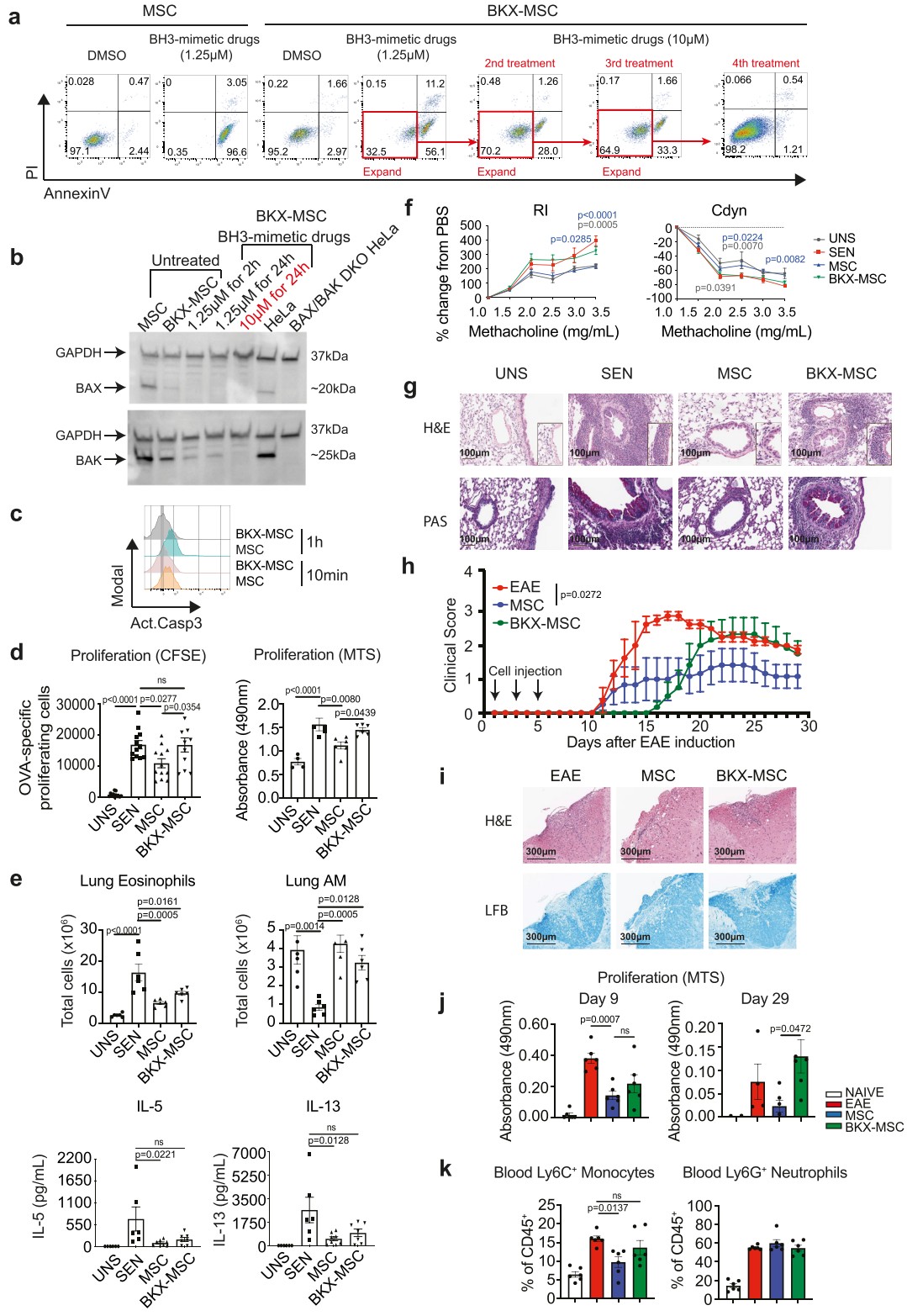

Supplementary Movie 1 and 2). Moreover, partial and transient in vivo depletion of AMs by intranasal administration of clodronate-loaded liposomes[24] prolonged the persistence of luciferase-expressing MSCs (Fig. 5F). These data obtained using a variety of approaches demonstrated in vivo efferocytosis of apoptotic MSCs in the lungs within 1 h of i.v. administration. Furthermore, we found evidence that AMs were a critical lung resident phagocytic population in MSC clearance that underwent phenotypic changes following efferocytosis.

**MSC treatment induces an anti-inflammatory transcriptional programme involving immunometabolic changes in AMs.** Our findings that AMs are critical for the therapeutic effect of MSCs[24], combined with their efferocytosis of these cells, prompted us to

**Fig. 3 BKX-MSCs display reduced immunosuppressive capacity in vivo. a** Generation of BKX-MSCs by selective expansion of apoptosis-resistant MSCs following multiple rounds of treatment with BH3-mimetic drugs (1.25 μM x 2 rounds, then 10 μM x 2 rounds). **b** Nontargeted MSCs and BKX-MSCs were treated with 1.25 μM or 10 μM BH3-mimetic drugs, for 2 or 24 h, then lysed and analysed for BAX and BAK protein levels by Western blotting. GAPDH served as the control for equal protein loading. Untreated MSCs and HeLa cells served as positive controls, and BAX/BAK double knockout (DKO) HeLa cells served as negative control. Data representative of two independent experiments. **c** Activated caspase-3 levels in CTV-labelled MSCs or BKX-MSCs re-isolated from mouse lungs 10 min and 1 h after i.v. injection. Data representative of two independent experiments. **d** OVA-sensitised mice received MSCs or BKX-MSCs prior to OVA challenge. OVA-specific DLN cell proliferation, measured by CFSE dilution (UNS $n = 12$; SEN $n = 12$; MSC $n = 12$; BKX-MSC $n = 11$) and MTS bioreduction (UNS $n = 4$; SEN $n = 5$; MSC $n = 6$; BKX-MSC $n = 6$). Data expressed mean ± SEM. $p$ values by one-way ANOVA (Tukey's post hoc test); ns not significant. **e** Number of eosinophils ($n = 5$ mice per group) and AMs ($n = 6$ mice per group) in the lungs, and DLN OVA-specific IL-5 and IL-13 production (UNS $n = 6$; SEN $n = 6$; MSC $n = 8$; BKX-MSC $n = 7$). Data expressed as mean ± SEM, $p$ values by one-way ANOVA (Tukey's post hoc test); ns, not significant. **f** Measurement of RI (UNS $n = 5$; SEN $n = 9$; MSC $n = 12$; BKX-MSC $n = 10$) and Cdyn (UNS $n = 4$; SEN $n = 9$; MSC $n = 12$; BKX-MSC $n = 10$) in response to increasing doses of methacholine on Day 12. UNS = unsensitized mice; SEN = OVA-sensitised mice; MSC = OVA-sensitised mice that received MSCs; BKX-MSC = OVA-sensitised mice that received BKX-MSCs. Data expressed mean ± SEM, $p$ values by two-way ANOVA (Tukey's post hoc test), compared with SEN. **g** Lung sections were stained with H&E and PAS to analyse for pulmonary inflammation and mucus production, respectively. Magnification 20x, scale bar = 100 μm. Histological images were representative of 5 mice per group. **h** Mean daily clinical scores of EAE mice that received PBS, MSCs or BKX-MSCs on Days 1, 3 and 5 (arrows indicate days of MSC injections). Data expressed as mean ± SEM, six mice per group. $p$ value by Kruskal-Wallis (Dunn's post hoc test). **i** CNS sections were stained with H&E and LFB to assess infiltrating inflammatory cells and demyelination respectively. Representative of 6 mice per group. Scale bar = 300 μm. **j** MOG-specific LN cell proliferation of EAE mice that received PBS, MSCs or BKX-MSCs and euthanised on Day 9 ($n = 6$ mice per group) and 29 (NAÏVE $n = 3$; EAE $n = 4$; MSC $n = 6$; BKX-MSC $n = 6$). Data expressed as mean ± SEM, $p$ values compared to PBS group (one-way ANOVA, Tukey's post hoc test). **k** Percentage of CD45$^+$Ly6G$^-$Ly6C$^{hi}$ monocytes and CD45$^+$Ly6G$^+$Ly6C$^{int}$ neutrophils in the blood on Day 9. Data expressed as mean ± SEM, $n = 6$ mice per group. $p$ values by one-way ANOVA (Tukey's post hoc test). Source data are provided as a Source Data file.

investigate how MSC treatment influenced AMs in the context of allergic asthma. AMs were isolated prior to (D8) and following (D12) intranasal allergen challenge for RNA sequencing. We first mapped the reads to the human genome to confirm that there was no significant contribution from the human MSCs. Principal component analysis (PCA) of differentially expressed genes (DEGs) showed relatively small, but discrete, transcriptional changes among the populations at D8 (Fig. 6a). Greater transcriptional changes were apparent among populations from D12 (Fig. 6a). Approximately 80% of variance was due to Dimension 1, which distinguished AMs from OVA-sensitised mice (SEN) versus unsensitised mice (UNS), while Dimension 2 appeared to distinguish AMs from MSC-treated mice (SEN + MSC and UNS + MSC) versus untreated mice (UNS and SEN) (Fig. 6a). AMs from MSC-treated OVA-sensitised mice (SEN + MSC) clustered more closely with those from unsensitised mice (UNS) than untreated OVA-sensitised mice (SEN), suggesting the presence of a therapeutic signature.

We found 422 genes to be differentially expressed between SEN mice and SEN + MSC mice, while 841 genes were associated with asthma and not associated with MSC administration (Fig. 6b). Strikingly, KEGG pathway analysis of the 422 DEGs revealed the Phagosome pathway to be the top significantly enriched pathway (Fig. 6c), in line with our findings and other studies indicating that the immunomodulatory effects of MSC therapy are due in part to the efferocytosis of MSCs[11,18].

Macrophages are notoriously plastic, adapting to their microenvironment to become broadly M1 or M2 macrophages with differing effector function[33]. Cognisant of the context-dependent activation and polarisation of macrophages[34], we undertook an analysis of genes broadly associated with M1/M2 macrophage polarisation to assess how MSC treatment modifies AM phenotype. Comparison of DEGs from UNS and SEN mice confirmed that AMs express genes associated with the M2 phenotype in allergic asthma (Fig. 6d)[24], likely because the allergic lung environment is enriched in IL-4 and IL-13, which drive M2 polarisation[35]. These M2-associated genes were downregulated following MSC administration (Figs. 6d, 6e). These genes include *Ccl17* and *Ccl22*, which are ligands for CCR4 pathway mediating allergen-induced recruitment of T cells to the airway[36,37], *Ccl24*, which induces eosinophil chemotaxis[38], and

*Arg1*, which is involved in l-arginine metabolism underlying airway hyperresponsiveness[39].

Genes encoding other proteins known to drive lung inflammation or asthma were also downregulated in MSC-treated mice (Fig. 6e). Notable genes include *IL31ra*, which encodes the receptor for IL-31 implicated in allergic asthma pathogenesis[40], and *Timp1*, which regulates matrix metalloproteinases implicated in airway remodelling[41] (Fig. 6e). On the other hand, the M1-associated genes, *Il1β*, *Il18*, *Ccl9* and *Cd86*, showed modest changes in SEN mice compared to UNS mice, with AM profiles reverting to near basal (i.e. unsensitised) levels when MSCs were administered to the SEN mice (Fig. 6d). Overall, our data support a scenario where MSC treatment antagonises M2-polarisation of AMs in asthma.

Of the DEGs that were upregulated in AMs from SEN + MSC mice, a notable gene is *Il10ra* (Fig. 6e). *Il10ra* encodes the receptor for interleukin-10 (IL-10) and mediates its immunosuppressive signal in IL-10-producing macrophages induced by MSC treatment in septic mice[23]. Another highly upregulated gene is *Cd300e* (Fig. 6e), a member of the CD300 receptor family with emerging roles in myeloid cell efferocytosis[42]. Interestingly, a striking preponderance of upregulated DEGs were type I/II IFN-responsive genes (e.g. *Il10ra*, *Ly6e*, *Gbp3*, *Slfn1*, *Slfn8*, *Stat1*, *Lgals3bp*, *Cxcl9* and *Cybb*) (Fig.6e). Gene set enrichment analysis (GSEA) against 50 hallmark gene sets confirmed a significant enrichment of genes from the interferon-alpha and interferon-gamma response sets, which were the top 2 out of 9 gene sets that were upregulated in SEN + MSC (Fig. 6f). GSEA also identified 16 gene sets that were downregulated in SEN + MSC, with an enrichment of genes involved in metabolism, including oxidative phosphorylation, fatty acid metabolism and glycolysis (Fig. 6f).

Thus, efferocytosis of apoptotic MSCs induces IFN-responsive genes and metabolic reprogramming in AMs to inhibit lung inflammation.

**AMs from MSC-treated mice are immunosuppressive.** Our data suggest that the inhibitory effects of MSC administration are mediated by the host phagocytic response to MSC apoptosis, rather than cell-intrinsic immunosuppression by viable MSCs. We also found that apoptotic MSCs did not directly inhibit T cell proliferation (Fig. 7a). We therefore investigated whether lung

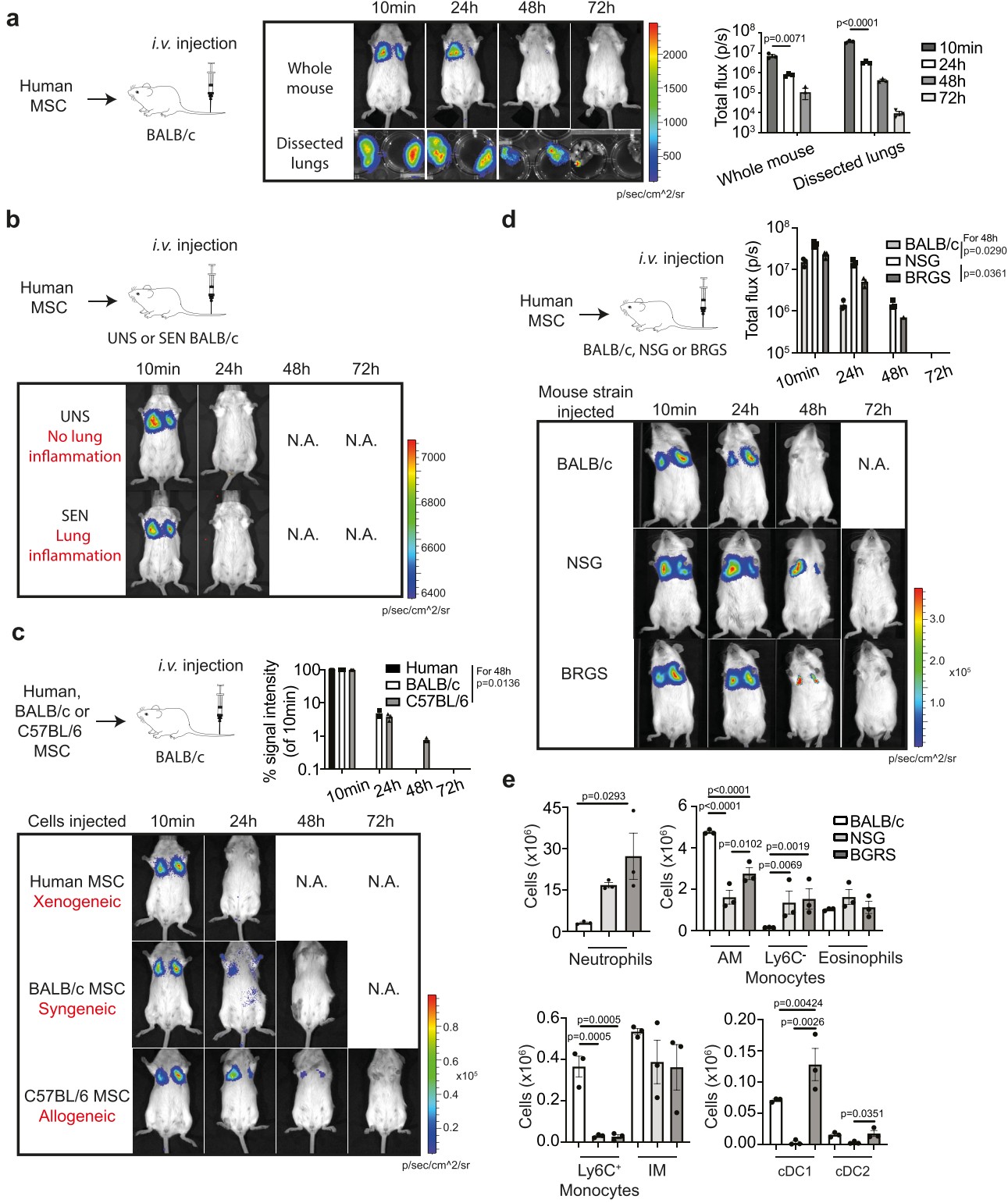

phagocytes from MSC-treated mice could directly inhibit the T cell response to allergen restimulation. DLN cells from OVA-sensitised mice were less proliferative when restimulated with OVA in the presence of AMs from MSC-treated mice compared to AMs from mice that did not receive MSCs (Fig. 7b). Neutrophils and monocytes from MSC-treated mice, on the other hand, did not inhibit the proliferative response to OVA (Fig. 7b).

Given our finding that *Il10ra* was upregulated in AMs from MSC-treated mice (Fig. 6c), we investigated whether this receptor

upregulation was in response to an increase in IL-10. AMs, but not neutrophils or monocytes, from MSC-treated mice induced an increase in IL-10 production by DLN cells from OVA-sensitised mice upon restimulation with OVA (Fig. 7c).

To investigate the longevity of MSC-induced immunomodulatory effects, we administered MSCs in an allergen re-exposure model that recapitulated characteristic features in allergic asthma (Fig. 7d). In this model, OVA-sensitised mice were rested for a week before a second round of allergen exposure. MSC-treated

**Fig. 4 MSCs are cleared despite the absence of cytotoxic cells or allorecognition. a** Bioluminescent images of the whole body and dissected lungs, and quantification of total flux following i.v. administration of human MSCs into BALB/c mice. Data expressed as mean ± SEM, $n = 3$ mice per group, representative of two independent experiments. $p$ values by two-way ANOVA (Tukey's post hoc test). **b** Bioluminescent images of unsensitised (UNS) versus OVA-sensitised (SEN) mice injected with human MSCs following intranasal challenge with OVA. Representative images of two independent experiments ($n = 3$ mice per group) **c** Administration of human MSCs (xenogeneic) versus BALB/c MSCs (syngeneic) versus C57BL/6 MSCs (allogeneic) into BALB/c mice. Bioluminescent images and quantification of total flux with data expressed as mean ± SEM, three mice per group, representative of two independent experiments. $p$ values for 48 h by two-way ANOVA (Tukey's post hoc test). **d** Administration of human MSCs into BALB/c versus NSG versus BRGS mice. Bioluminescent images and quantification of total flux with data expressed as mean ± SEM, $n = 3$ mice per group, representative of two independent experiments. $p$ values for 48 h by two-way ANOVA (Tukey's post hoc test). **e** Total cell number of various myeloid cell populations measured in lungs of BALB/c, NSG and BGRS mice. Data expressed as mean ± SEM, $n = 3$ mice per group, representative of two independent experiments. $p$ values by one-way ANOVA (Tukey's post hoc test). Source data are provided as a Source Data file.

mice exhibited decreased AHR, characterised by decreased Rl and increased Cdyn in response to increasing doses of methacholine, compared to OVA-sensitised mice that did not receive MSCs (Fig. 7e). MSC-treated mice displayed decreased BALF eosinophils, T cell proliferative response to OVA, and IL-5 and IL-13 production (Fig. 7f). Lung tissue sections also showed a reduction in inflammatory cells and mucus-secreting goblet cells in the airways of MSC-treated mice (Fig. 7g). The inhibitory effects of MSC administration on eosinophils persisted (albeit abating) when OVA-sensitised mice were rested for 4 weeks before allergen re-exposure (Fig. 7g). Similarly, when MSCs were administered after allergen exposure, OVA-sensitised mice exhibited decreased eosinophil influx upon re-exposure to an allergen (Fig. 7h). Thus, disease parameters were inhibited several weeks after MSCs were cleared from the lungs.

Taken together, our data demonstrate that efferocytosis of apoptotic MSCs causes sustained alterations in AM immunometabolism and function that directly inhibit lung inflammation.

## Discussion

The current consensus is that the MSC secretome is their main mode of action. This mechanism requires secreted factors to act across long distances with effects that last beyond the short half-life of MSCs. To our knowledge, these features have not yet to be clearly demonstrated. By contrast, recent studies have linked MSC apoptosis with their therapeutic effects in animal models of GvHD[11], sepsis[17], acute lung injury[16] and allergic airway inflammation[18]. However, as these studies did not investigate the clinical impact of impairing MSC apoptosis, the relative contribution of known physiological effectors of viable MSCs versus the host response to dying MSCs remains unclear. We demonstrate here that impairing apoptosis in MSCs reduces their immunomodulatory capacity in vivo, thereby establishing that apoptosis is necessary for the full beneficial effects of MSC therapy.

MSCs are most commonly administered via the i.v. route[43,44] but encounter a pulmonary impasse. Cell death of MSCs has been reported using bioluminescence[11], cryo-imaging[10] or microarray analysis[7], which may not detect small numbers of surviving MSCs. Using highly sensitive flow-based approaches to detect MSCs on a per-cell basis, we show that virtually all MSCs became apoptotic, a process that is initiated within 1 h of i.v. administration. MSCs were labelled with a cytoplasmic dye (CTV), which tracks cells with higher fidelity than membrane-bound dyes (e.g. PKH) that are retained even when cells lose their plasma membrane integrity. Thus, it is highly unlikely that small numbers of MSCs survive and contribute to immunosuppression in the lungs. We further provide genetic proof for mitochondrial apoptosis as the predominant cell death pathway in MSCs lodged in the lungs, as the ablation of BAK/BAX expression protected MSCs from the rapid induction of apoptosis. Importantly, BAK/BAX-deficient MSCs were unable to inhibit asthma and EAE to the full extent as

nontargeted MSCs, indicating that a key cell-intrinsic property of MSCs as a therapeutic agent is their apoptosis. This is further supported by the ineffectiveness of MEFs, a cell type that is relatively resistant to apoptosis induction via BH3-mimetic drugs. Whilst it is not unexpected that apoptotic MSCs would be taken up by a range of haematopoietic cell types with phagocytic capacity, a key finding is that this process induces transcriptional changes that alter immunometabolism in AMs, drive IL-10 responsiveness and blunt inflammatory responses, leading to durable immunosuppression. Our study unifies these aspects of MSC therapy to provide clarity around their mechanism of action.

Although our study is limited to i.v. administration, the impact of MSC apoptosis in the lung extends to a disease setting where tissue damage occurs at a site distant to the lung. In EAE, the lung serves as a homing niche and activation site through which myelin-reactive T cells must traffic before entering the CNS and causing damage[45]. A role for lung immune cells in EAE is further supported by the successful treatment of EAE using approaches that deliver soluble antigens to the lung[46]. The EAE model allowed us to establish a connection between MSC apoptosis in the lung and peripheral immune responses that subsequently impact remote tissue damage. Our EAE data support a scenario where impeding MSC apoptosis in the lung alters host immune responses that delay (but not prevent) the clinical manifestation of disease.

The activation of caspase 3 in MSCs excludes the possibility that MSCs undergo inflammatory cell death (e.g. necroptosis), which is caspase-independent[13]. Using BH3-mimetics to target specific BCL-2 family members, our data showed that the administration of apoptotic MSCs was sufficient to mediate immunosuppression in the lungs. This finding supported previous studies that demonstrated the efficacy of serum-deprived apoptotic MSCs in reducing disease parameters in rat models of bleomycin-induced[16] or sepsis-induced[17] lung injury. However, it is possible that apoptotic MSCs are less potent in other disease settings[47], as shown in a GvHD mouse model where apoptotic MSCs induced via FAS stimulation were only effective when administered i.p. but not via the i.v. route[11]. The efficacy of apoptotic MSCs may also be dependent on the stage of cell death at the time of administration, as heat-killed or PFA-fixed MSCs were not effective[23,24,44,48,49]. In this regard, MSCs that can actively undergo apoptosis may be regarded as "fit" compared to MSCs that are already dead or fixed[44].

Our finding that MSC apoptosis was not dependent on allorecognition and could occur in the absence of MHC mismatch suggests that MSCs would undergo the same fate regardless of whether cells were patient-derived or off-the-shelf. Furthermore, human MSCs were similarly cleared in NOD/SCID/$Il2r\gamma c^{-/-}$ mice and NOD.$sirpa$ $Rag2^{-/-}$ $Il2r\gamma c^{-/-}$ mice lacking T cells, B cells and NK cells, in accordance with the clearance of syngeneic mouse MSCs in $Rag2^{-/-}$ $Il2r\gamma c^{-/-}$ mice[8]. Thus, MSC killing by

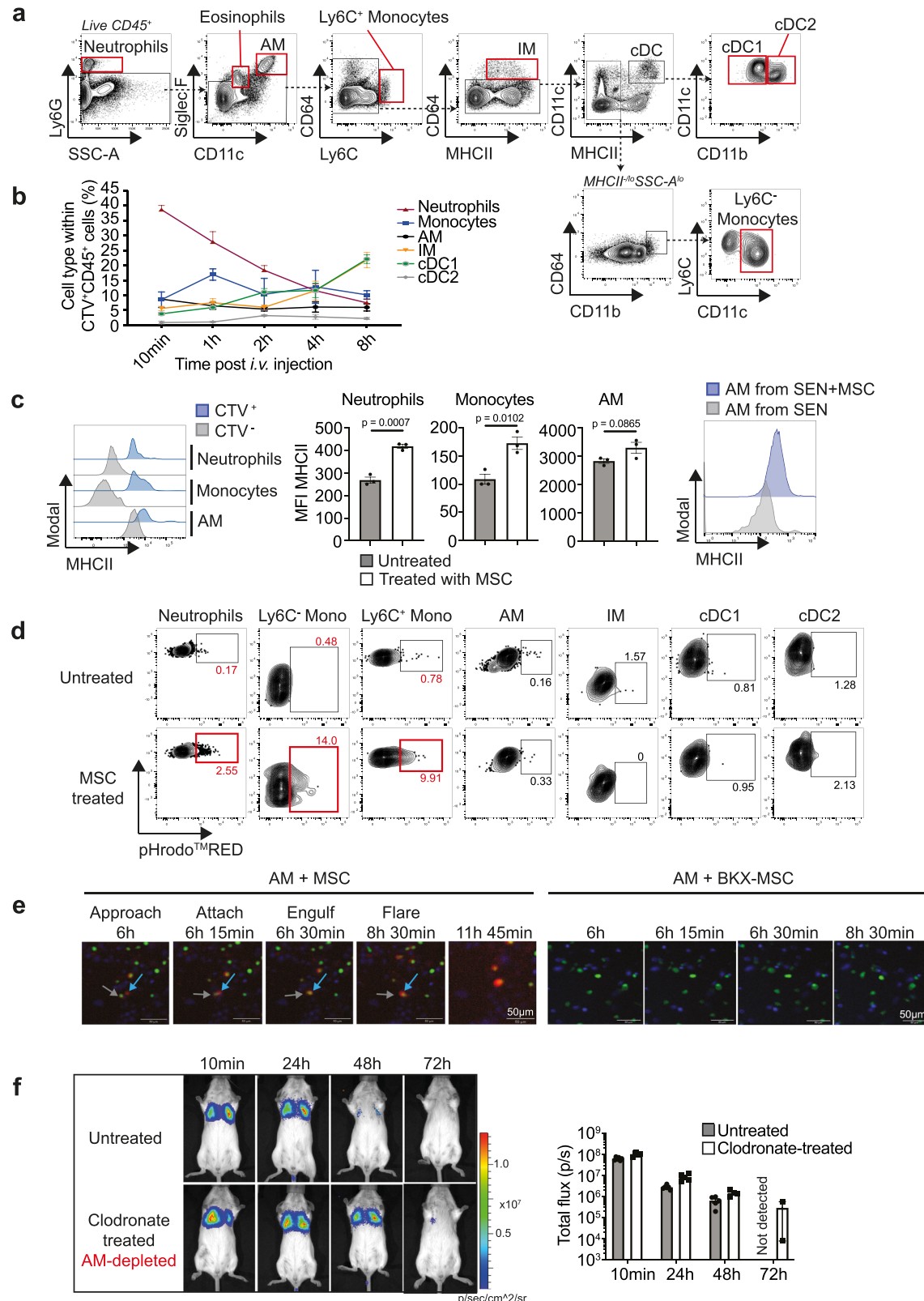

cytotoxic T cells and FAS ligation, as reported for GvHD[11], is not the main mechanism of MSC death in disease settings that do not involve an influx of allogeneic T cells into the lungs. This is further supported by a lack of caspase 3 activation in BAK/BAX-deficient MSCs at a time postinjection when nontargeted MSCs were already apoptotic, indicating a limited role for cytotoxic killing or triggering via the extrinsic pathway at this early time

point. Likewise, complement activation, shown to damage MSCs and reduce their viability[50], is unlikely to be the main mechanism, as NOD/SCID/*Il2rγc*−/− mice are deficient in haemolytic complement[51]. Furthermore, treatment of MSCs with mouse serum did not induce apoptosis at 1 hr, suggesting a limited role for complement killing at this early time point when i.v. injected MSCs were already apoptotic in the lungs. Our data also showed

**Fig. 5 MSCs are taken up by phagocytic cells in the lungs. a** Flow cytometric gating strategy for identifying myeloid cell populations in the lungs. **b** Frequencies of lung phagocytes that had taken up CTV-labelled MSCs at various timepoints after i.v. injection into BALB/c mice. Data expressed as mean ± SEM, n = 3 mice per group, three independent experiments. **c** Expression levels and MFI of MHC class II in neutrophils, monocytes and AMs that had taken up CTV[+] MSCs 1 h after injection, compared to phagocytes that had not (CTV[−]). Data expressed as mean ± SEM, n = 3 mice per group, three independent experiments. p values by unpaired Student's t test (two-tailed). Right-most panel shows MHC class II expression in AMs isolated from the lungs of OVA-sensitised mice on Day 12. SEN = OVA-sensitised mice; SEN + MSC = OVA-sensitised mice that received MSCs. Data representative of three independent experiments. **d** Flow cytometric plots of various phagocyte populations in the lungs, following injection of pHrodo[TM]RED-labelled MSCs. Data representative of two independent experiments; n = 3 mice per group. **e** Snapshots from live-cell imaging, showing CTG-labelled AMs (grey arrow) approach, attach and engulf pHrodo[TM]RED-labelled BH3-mimetic drug-treated MSCs (blue arrow), resulting in a signal flare (Supplementary Movie 1). BH3-mimetic drug-treated BKX-MSCs were not engulfed by AMs (Supplementary Movie 2). Magnification, 10x; Scale bars, 50 μM. Data representative of three experiments. **f** Bioluminescent images and total flux at various timepoints following injection of MSCs into untreated (n = 6) and clodronate-treated (AM-depleted; n = 5) BALB/c mice. Data expressed as mean ± SEM, 5–6 mice per group. Source data are provided as a Source Data file.

that local inflammation did not prolong the persistence of MSCs in the lungs, supporting studies that reported a lack of MSC persistence in models of acute lung injury[52,53]. Taken together, our data suggest that whilst immune cell-related cytotoxicity or complement may play a role in the blood or other settings, MSC apoptosis in the lung is likely induced by other factors, such as nutrient/growth factor deprivation or mechanical stressors.

Apoptotic cells can release immunosuppressive factors and extracellular vesicles (including apoptotic bodies) that modify various aspects of immune function[54]. They can also indirectly modulate the function of immune cells upon engulfment, whereby the induction of immunomodulatory macrophages, tolerogenic DCs, and regulatory T and B cells leads to the production of immunosuppressive cytokines, such as TGF-β and IL-10[55]. We and others have previously shown that MSC treatment in various settings induces regulatory T cells and IL-10[23,24,48], which may reflect the MSC apoptosis/efferocytosis axis. Apoptotic MSCs by themselves were not directly immunosuppressive (Fig. 7a). Instead, their efferocytic clearance is an important element of the mechanism. Neutrophils and monocytes were the major phagocytic cell types responsible for efferocytic clearance of MSCs (Fig. 5b), supporting previous findings[10]. However, the immunosuppressive effects of MSCs were primarily mediated by AMs in the lungs. Using OVA as a model antigen/allergen, AMs from MSC-treated mice directly suppressed antigen-specific proliferating cells while their monocytic counterparts did not. RNA-sequencing data indicated that MSCs had minimal effects on AMs prior to the onset of asthma, suggesting that inflammation induced by intranasal allergen challenge was required to initiate major transcriptional changes in AMs. The upregulation of Il10ra suggests a possible mechanism whereby AMs respond to increased IL-10 in the lungs of MSC-treated mice. Importantly, AMs but not neutrophils or monocytes from MSC-treated mice directly induced an increased production of IL-10 from allergen-responsive cells (Fig. 7c). This mechanism likely differs according to the disease setting and also the route of administration. For example, IDO was upregulated in mesenteric lymph node phagocytes that had engulfed i.p. injected apoptotic MSCs, but not in lung phagocytes that had engulfed i.v. injected MSCs[11]. Similarly, our RNA-sequencing data did not show IDO upregulation in AMs from MSC-treated mice.

Macrophage activation occurs on a continuum, and M1/M2 marker expression is plastic across different disease stages[34]. It may be too simplistic to assert that MSC treatment skews M1/M2 expression. Rather, our data indicate that MSC treatment decreases the expression of AM genes known to drive asthma and lung inflammation. These genes are mostly M2, but some are M1 (Fig. 6d). Thus, our data suggest a similar anti-inflammatory effect in other inflammatory environments with M1-primed macrophages, as shown in models of acute lung injury, sepsis and GvHD[11,16–18,23].

The significance of the upregulation of type I/II IFN-responsive genes in AMs remains to be determined. Interferons have been shown to induce Axl, which is critical for efferocytosis in AMs[56,57]. Recently, it was demonstrated in a model of myocardial infarction that cell therapy induces an acute inflammatory response in macrophages that stimulates the intrinsic healing cascade, a phenomenon that occurs also with cellular debris or when pattern recognition receptors are activated[58]. The induction of IFN-responsive genes could indicate such an acute inflammatory response. Furthermore, MSCs were found not to be protective in mice lacking the receptor for IFN-γ, suggesting that MSC effects were dependent on the recipient's ability to respond to IFN-γ[59]. Our data indicate that AMs are a critical cell type that can respond to IFN-γ. The upregulation of MHC class II on AMs further supports this interaction between AMs and IFN-γ, which is an inducer of MHC class II expression on mononuclear phagocytes[60]. Interferons have also been associated with metabolic reprogramming, a process that is intimately involved in regulating macrophage function[61]. A recent study showed that apoptotic cells release a range of metabolites that alter gene expression programmes in phagocytes, including pathways associated with inflammation and metabolism[62]. Whether the metabolic adaptations observed in AMs from MSC-treated mice reflect a similar apoptotic metabolite secretome is the focus of our current research.

The finding that MSC apoptosis and phagocytosis is pivotal in their therapeutic efficacy is consistent with the evidence supporting the prevailing "MSC licensing" concept. Many of the stimuli reported to license MSCs (e.g. IFN-γ, TNF-α and toll-like receptor activation) can also induce apoptosis[63–65]. Co-culture of MSCs and macrophages in the presence of LPS greatly enhanced phagocytosis of MSCs[66] and increased macrophage production of anti-inflammatory IL-10[23]. IFN-γ-activated MSCs also produced CCL2, crucial for recruiting IL-10-producing immunosuppressive monocytes/macrophages to the lungs[67]. Similarly, different inflammatory microenvironments can differentially alter MSC function by inducing apoptosis, increasing the expression of anti-inflammatory mediators and enhancing the phagocytic capacity of macrophages[68].

Immunosuppression has been demonstrated for apoptotic cells that are not MSCs, including apoptotic neutrophils[19], primary human T cells[14,15] and Jurkat T cells[20], via similar mechanisms involving efferocytosis. An outstanding question is whether the engulfment of MSCs induces specific changes in phagocytes that underlie the beneficial effects of MSC therapy, or whether similar effects could be exerted by other cell types undergoing apoptosis. Furthermore, the demonstration that viable MSCs are not necessary raises questions about the validity of potency assays that measure factors secreted or expressed by viable MSCs. Whilst soluble factors secreted by viable MSCs into the culture medium (e.g. extracellular

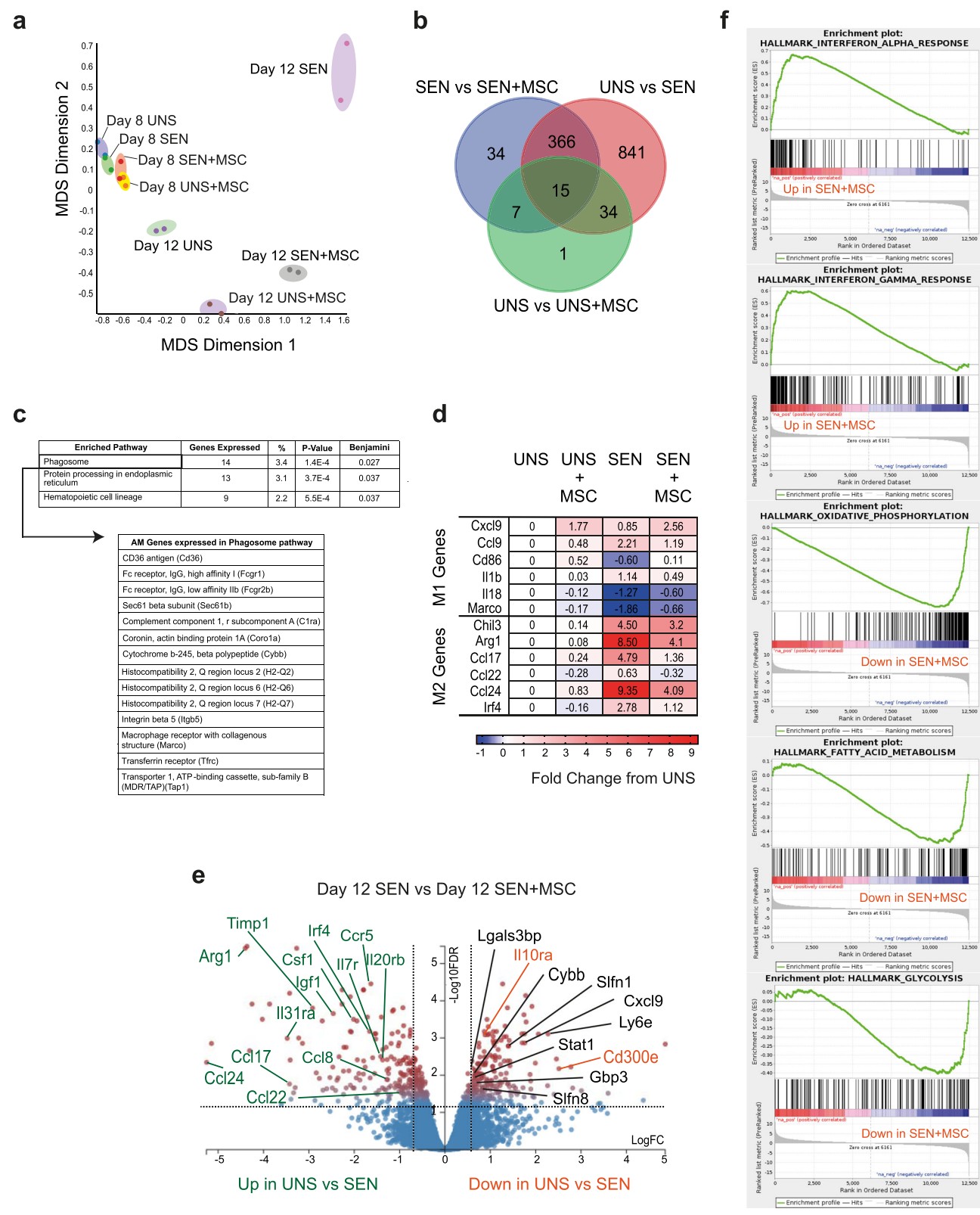

vesicles) offer an attractive cell-free therapeutic alternative, they may not be a direct measure of potency when viable MSCs are infused as a cellular product. On the other hand, apoptotic MSC-derived extracellular vesicles have recently been shown to demonstrate therapeutic potential[47,69–71]. Our data provide further understanding of the mechanisms of MSC therapy and reveal opportunities for novel therapeutic interventions.

## Methods

**Study design.** This study aimed to determine how the apoptosis of intravenously administered MSCs contributes to their therapeutic efficacy. In genetic loss-of-function experiments, we selectively blocked the intrinsic pathway of apoptosis in human MSCs to test whether this would abrogate their immunomodulatory effects in diseases in which MSCs are currently being trialled as therapeutic agents. We used a mouse model of asthma because (i) the OVA-induced asthma model uses fully immunocompetent mice and has been extensively characterised; and (ii)

**Fig. 6 MSC treatment causes major transcriptional changes in AMs to dampen lung inflammation and asthma. a** Unsensitised or OVA-sensitised mice received MSCs on day (D) 5, 6 and 7. AMs from BALF were FACS purified for RNA-sequencing either prior to (D8) or following (D12) OVA challenge. PCA analysis of genes from D8 and D12 treatment groups. Plots were generated using the top 100 variable genes within samples. Each dot represents data from 5 pooled samples. **b** Venn diagram showing number of overlapping DEGs among different treatment groups in D12, obtained by setting a filter of 1 CPM (counts per millions) and FDR of 0.05. **c** Classification of DEGs (obtained from **b**) by KEGG pathway enrichment. **d** Heatmap showing genes with an FDR of 0.05 that are broadly associated with M1/M2 macrophage polarisation. **e** Volcano plot comparing DEGs from D12 SEN versus D12 SEN + MSC groups. Gene expression changes with a fold-change greater or less than 1.5 were shown as red dots. Selected genes highlighted in green represent genes that were upregulated in UNS versus SEN and downregulated in SEN + MSC, whereas those in orange represent genes that were downregulated in UNS versus SEN and upregulated in SEN + MSC. Selected genes in black indicate interferon regulated genes identified via Interferome. **f** GSEA plots showing enrichment of genes from the Interferon-Alpha Response (ES: 0.66824204, NES: 2.684128, FDR $q$ value: 0.0, nominal $p$ value: 0.0), Interferon-Gamma Response (ES: 0.59893006, NES: 2.6195564, FDR $q$ value: 0.0, nominal $p$ value: 0.0), Oxidative Phosphorylation (ES: −0.74083024, NES: −3.1281621, FDR $q$ value: 0.0, nominal $p$ value: 0.0), Fatty Acid Metabolism (ES: −0.48348597, NES: −1.9388303, FDR $q$ value: 1.6666666E-4, nominal $p$ value: 0.0), and Glycolysis (ES: −0.4049271, NES: −1.6384116, FDR $q$ value: 0.013858369, nominal $p$ value: 0.0016447369) hallmark gene sets in D12 SEN + MSC compared to D12 SEN. Unbiased GSEA was performed using software from the Broad Institute[75] against hallmark gene sets ($n = 50$) and C2 curated gene sets ($n = 4063$) from the Molecular Signatures Database (MSigDB). Significant enrichment was defined as a $p \leq 0.05$ and FDR < 0.25. Interferon regulated genes were identified using the Interferome v2.01 database[76]. Source data are provided as a Source Data file.

inflammation occurs primarily in the lungs where injected MSCs localised to, and therefore has been used by many groups to assay in vivo human MSC efficacy. We used a mouse model of multiple sclerosis because MOG-induced EAE mimics the autoimmune destruction of the central nervous system and therefore enables us to measure the distal effects of MSCs undergoing apoptosis in the lungs. In all experiments, animals were randomly allocated to control or experimental groups. In the EAE experiments, disease scoring was performed blind and the endpoint was reached when untreated EAE mice reached a clinical score of 3.5, as per animal ethics committee recommendations. Unless otherwise specified, two to three independent experimental replicates were performed.

**Animals**. Female 7- to 9-week old BALB/c and C57BL/6 mice were obtained from Monash Animal Services and maintained under specific pathogen-free conditions at the Monash University Animal Research Laboratories. All animal experiments were conducted in accordance with the guidelines of the Australian Code of Practice for the Care and Use of Animals for Scientific Purposes and approved by the Monash University Animal Ethics committee (MARP/2016/160).

**OVA-induced allergic asthma model**. Female 7-week-old BALB/c mice were sensitised with 50 μg chicken ovalbumin (OVA, grade V; Sigma-Aldrich, USA) and 2 mg aluminium hydroxide (SERVA; Thermo Fisher, USA) in Dulbecco's PBS (Invitrogen, USA), administered via intraperitoneal (i.p.) injection on day 0. Control mice received 2 mg aluminium hydroxide in Dulbecco's PBS. To examine the effects of MSC administration upon allergen challenge, mice received $1 \times 10^6$ MSCs, STS-MSCs, BH3-MSCs or BKX-MSCs in 200 μL Dulbecco's PBS via tail vein injections on days 5, 6 and 7. Mice were then challenged with 50 μg OVA administered intranasally under light anaesthesia on days 8, 9, 10 and 11 (otherwise indicated). On day 12, lung function measurements were performed, and tissues were harvested for subsequent analyses. To examine the longevity of MSC treatment effects, mice were re-challenged daily with intranasal OVA on days 18–21 and analysed on day 22, or re-challenged on days 38–41 and analysed on day 42. To recapitulate established asthma, OVA-sensitised mice were challenged with intranasal OVA on days 8, 9 and 10, injected with MSCs on days 15, 16 and 17, then re-challenged with intranasal OVA on day 18–21 and analysed on day 22.

**MOG-induced EAE model**. EAE was induced in 10- to 12-week-old, female C57Bl/6 mice by subcutaneous injection of 100 μg MOG$_{35–55}$ peptide emulsified in complete Freund's adjuvant (Sigma-Aldrich) supplemented with 400 μg Mycobacterium tuberculosis (Difco) into both hind limb flanks. Mice received an intraperitoneal injection of 350 ng Pertussis toxin on day 0 and day 2. Mice were monitored daily and scored blind to avoid unconscious bias. Clinical scores were assigned according to an arbitrary clinical scale: 0, no detectable impairment; 0.5, partial loss of tail tone; 1, complete loss of tail tone; 1.5, complete loss of tail tone and difficulty righting; 2, complete loss of tail tone and weak hind limbs; 2.5, one paralysed hind limb; 3, complete paralysis of hind limbs; 3.5, complete paralysis of hind limbs and ascending paresis affecting the trunk region; 4, complete paralysis of hind limbs and paresis in forelimbs; and 5, moribund or deceased. Under recommendation of the animal ethics committee, mice were euthanised upon reaching a clinical score of 3.5. In some experiments, mice were euthanised on Day 9. Blood was collected via cardiac bleed into heparin-coated tubes, red cell lysed and processed for flow cytometric analysis.

**Lung function**. Airway hyperresponsiveness (AHR) was assessed using restrained invasive plethysmography 1 day after the final intranasal OVA challenge. Mice were anesthetised by i.p. injection of a mixture containing ketamine hydrochloride

(Parnell, Australia) and xylazine hydrochloride (Ilium Xylazil-20, Troy Laboratories, Australia) at a dose of 80–100 mg/kg and 10–13 mg/kg body weight, respectively. A small incision was made to expose the trachea, and a cannula was inserted to connect to an inline nebuliser and ventilator. Mice were challenged with aerosolised PBS followed by increasing doses of methacholine (Sigma-Aldrich). Airway resistance (RI) and dynamic compliance (Cdyn) were recorded and determined by analysis of pressure and flow waveforms with Buxco FinePointe software v3.2 (Data Sciences International, USA).

**Bronchoalveolar lavage and lung digestion**. Mice were euthanised by pentobarbitone overdose 2 days after the last intranasal OVA challenge. Bronchoalveolar lavage fluid (BALF) was obtained by instilling three washes of 0.4 ml PBS with 0.1% BSA. BALF was centrifuged at 470x $g_{max}$ for 5 min and the cell pellet were resuspended in FACS buffer (0.1% BSA and 2.5 mM EDTA in PBS), enumerated and labelled for flow cytometric analysis. Lungs were snipped into small fragments and digested for up to 1 h in lung digestion media (300 U/mL Collagenase type I (Worthington) and 50 U/mL DNAse I (Sigma-Aldrich) in RPMI-1640) in a 37 °C water bath with occasional agitation using a pipette. The digested lung samples were passed through a 70-micron cell strainer and centrifuged. The cell pellet was resuspended in red blood cell lysis buffer, washed, enumerated and resuspended in FACS buffer for subsequent flow cytometry analysis. Cell counting was performed using a Z2 Coulter Counter (Beckman Coulter).

**Flow cytometric analysis**. Single-cell suspensions were labelled with a panel of primary antibodies targeting CD45 (clone 30-F1; BD Biosciences), CD11b (clone M1/70; eBioscience), CD11c (clone HL3; BD Biosciences), Ly6C (clone HK1.4; Biolegend), Ly6G (clone 1A8; BD), CD24 (clone M1/69; BD Biosciences), CD64 (clone X54–5/7.1; Biolegend), MHC II (clone M5/114.15.2; Biolegend), and Siglec F (clone 1 RNM44N; eBioscience), CD115 (CSF-1R; clone AFS98; Biolegend), CD326 (EpCAM; clone G8.8a; BD Biosciences), CD104 (clone 346–11 A; Biolegend) and CD31 (clone MEC13.3; BD Biosciences). Streptavidin PE-Cy7 (Biolegend), Streptavidin APC-Cy7 (Biolegend) and Streptavidin BV510 (Biolegend) were used as secondary antibodies. FcR block (clone 2.4G2; WEHI) was added to reduce non-specific binding. Fixable Viability Stain (Invitrogen) was used to distinguish live cells from dead cells. For detection of intracellular markers, cells were fixed using Cytofix/Cytoperm solution (BD Biosciences) or eBioscience™ Foxp3 / Transcription Factor Staining Buffer Set (Invitrogen) after surface marker labelling. Apoptotic cells were detected using AnnexinV/propidium iodide (PI) or active caspase-3 (Act.Casp3) (BD Biosciences). Samples were acquired via a LSRFortessa X-20 cell analyser (BD Biosciences) with BD FACSDIVA (BD Biosciences v6.0) and analysed using FlowJo v10 software. Immune cell types in the lungs were gated as shown in Fig. 5a, according to[31] and[72]. All antibodies used in this study were tested using cells from spleens, bone marrows, lungs of mice, or human cell lines to determine optimal titration for staining.

To isolate alveolar macrophages, monocytes and neutrophils, lungs were digested to single-cell suspensions as described above. The cells were labelled with biotinylated antibody to CD45, followed by antibiotin magnetic bead (Militenyi Biotec). The cells were then subjected to magnetic separator, and the positive fraction containing CD45$^+$ cells were collected labelled with a panel of antibodies. Cell sorting was performed on a BD Influx™ Cell Sorter (BD Biosciences).

**Antigen-specific proliferation and cytokine production**. For OVA-specific proliferation, lung-draining mediastinal and bronchial lymph nodes (DLN) were harvested, gently mashed and washed with RP10 media (RPMI-1640 media with 10% heat-inactivated FBS) through a 70-micron cell strainer. DLN cells were plated at $5 \times 10^5$ cells/well in 96-well round-bottom plates in triplicate and stimulated

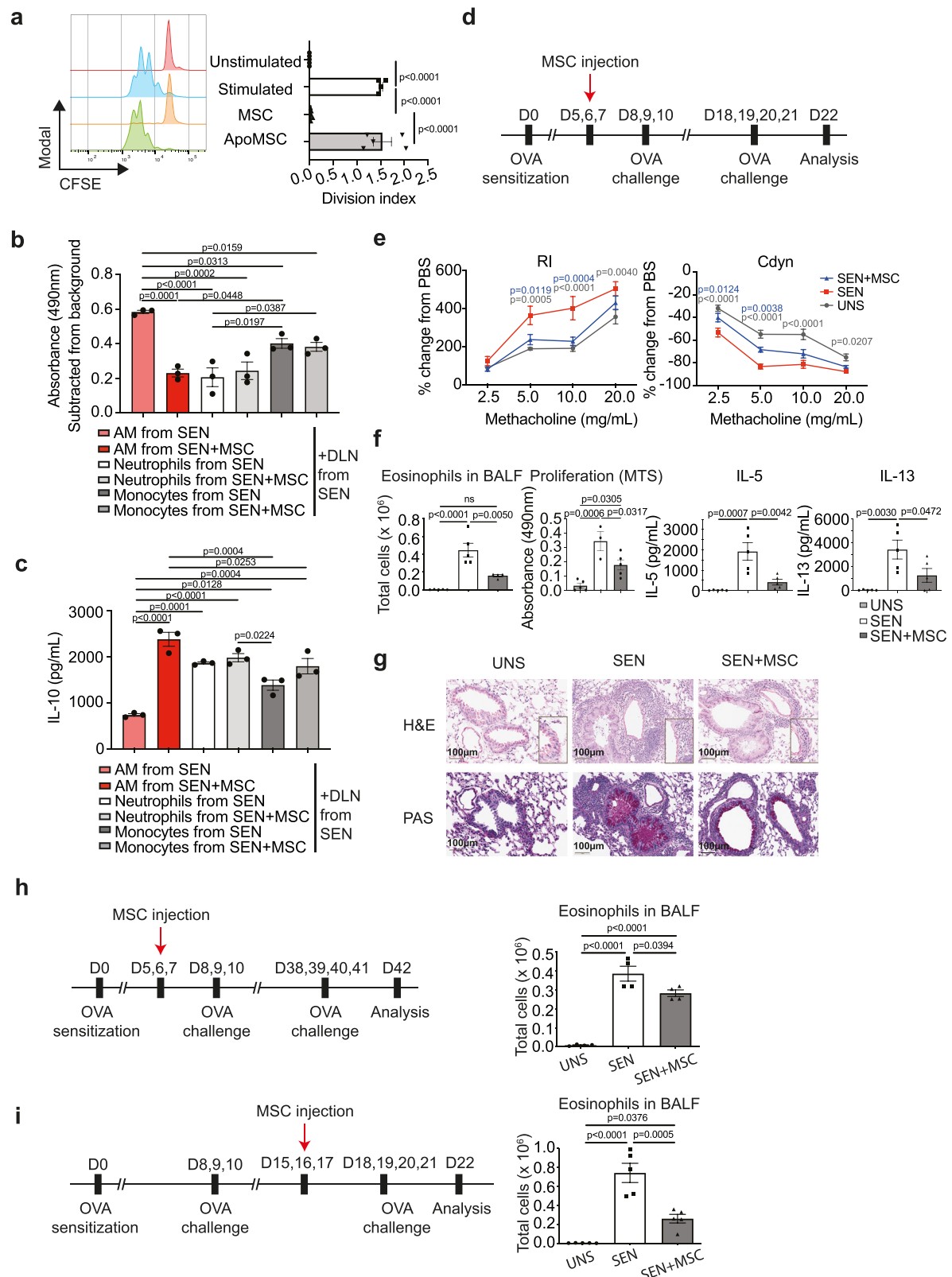

with 20 μg of OVA per well. Unstimulated control was cells cultured in RP10 without OVA. To analyse cell proliferation via CFSE dilution, DLN cells were labelled with CellTrace™ CFSE, according to manufacturer's protocol (Thermo-fisher), and stimulated with OVA for 5 days. In some experiments, cell proliferation was analysed on day 3 via MTS bioreduction using CellTiter 96® AQ$_{ueous}$ One Solution according to manufacturer's protocol (Promega). Background proliferation in the unstimulated wells was subtracted from the number of proliferating cells in the OVA-stimulated wells. For OVA-specific cytokine production, $4 \times 10^5$ DLN

cells/well were added to 96-well flat-bottom plates containing 20 μg OVA per well. The supernatant was collected after 4 days of culture and analysed for various cytokines, using either BD Cytometric Bead Array (BD Biosciences) or LEGEN-Dplex (Biolegend) according to the manufacturer's instructions. Data were collected on a FACSCanto II or LSRFortessa X-20 and analysed using FCAP Array software v3.0 (BD Biosciences) or LEGENDplex Data Analysis Software v8.0 (Biolegend). To assess the inhibitory effects of myeloid cells from MSC-treated mice, DLN cells from OVA-sensitised mice were plated at $2.5 \times 10^5$ cells/well or

**Fig. 7 AMs from MSC-treated mice exhibit immunomodulatory effects. a** Flow cytometric histogram plot and bar chart demonstrating that humans MSCs, but not apoptotic MSCs (ApoMSCs, BH3-mimetic drug-treated), inhibited the proliferation of αCD3/CD28-stimulated CFSE-labelled purified T cells (Unstimulated $n = 5$; Stimulated $n = 4$; MSC $n = 5$; ApoMSC $n = 5$). Data expressed as mean ± SEM, representative of two independent experiments. $p$ values by one-way ANOVA (Tukey's post hoc test). **b** Monocytes and AMs were FACSorted from SEN or SEN + MSC-treated mice on D12, then cocultured with DLN cells from SEN mice (1:5 myeloid cell:responder ratio) in the presence of OVA. AMs, but not monocytes, from SEN + MSC mice inhibited the proliferative response to OVA. Data expressed as mean ± SEM, representative of two independent experiments; $n = 3$ biological independent samples per group. $p$ values by one-way ANOVA (Tukey's post hoc test). **c** AMs, but not monocytes, from SEN + MSC mice induced IL-10 production in response to OVA. Data representative of two independent experiments. Data expressed as mean ± SEM, representative of two independent experiments, $n = 3$ biological independent samples per group. $p$ values by one-way ANOVA (Tukey's post hoc test). **d** Timeline for MSC treatment in Day 22 OVA re-challenge model. **e** RI and Cdyn of mice from **D** in response to increasing doses of methacholine. Data expressed as mean ± SEM (UNS $n = 7$; SEN $n = 6$, SEN + MSC $n = 8$). $p$ values by two-way ANOVA (Tukey's post hoc test), compared with SEN. **f** Number of eosinophils in BALF (UNS $n = 5$; SEN $n = 5$, SEN + MSC $n = 4$), OVA-specific DLN proliferation, measured by MTS bioreduction (UNS $n = 5$; SEN $n = 3$, SEN + MSC $n = 5$) and IL-5 and IL-13 production after OVA restimulation of T cells ($n = 5$ per group). Data expressed as mean ± SEM, $p$ values by one-way ANOVA (Tukey's post hoc test). **g** Lung sections were stained with H&E and PAS to analyse for pulmonary inflammation and mucus production respectively. Magnification 20x, scale bar = 100 μM. Histological images were representative of five mice per group. **h** Timeline for MSC treatment in Day 42 OVA re-challenge model and number of eosinophils in BALF. Data expressed as mean ± SEM ($n = 5$ per group). $p$ values by one-way ANOVA (Tukey's post hoc test). **i** Timeline for MSC treatment after intranasal OVA challenge and total number of eosinophils in BALF. Data expressed as mean ± SEM ($n = 5$ per group). $p$ values by one-way ANOVA (Tukey's post hoc test). Source data are provided as a Source Data file.

cocultured with $0.5 \times 10^5$ sorted myeloid cells (1:5 myeloid cell: responder ratio) in the presence of OVA for 3 days. MOG-specific assays were performed as described above by harvesting cervical and inguinal lymph nodes and stimulating with 20 μg MOG$_{35-55}$.

**In vitro suppression assay.** To isolate T cells, mouse splenocytes were labelled with biotinylated antibodies to CD11b, CD11c, CD19, B220, NK1.1, CD326, MHC II and Ter119, followed by antibiotin magnetic beads (Militenyi Biotec). Cells were passed through a LS column (Militenyi Biotec) on a QuadroMACS™ magnetic separator, and the negative fraction containing the untouched T cells was collected. T cells were labelled with CellTrace™ CFSE, according to manufacturer's protocol (Thermofisher). Ninety-six-well round-bottom plates were coated with 50 μL of 5 μg/mL purified NA/LE αCD3 (clone 145-2C11, BD Biosciences) for 2 h at 37 °C. The αCD3 was then removed and the wells were washed with PBS to remove excess antibody. Soluble αCD28 (clone 37.51, BD) was added at 2 μg/mL final concentration. Unstimulated wells contained RP10 media without αCD3/CD28. CFSE-labelled purified T cells were plated at $5 \times 10^5$ cells/well or cocultured with $1 \times 10^5$ MSCs (1:5 MSC:responder ratio). Cells were harvested after 3 days for analysis by flow cytometry. Cells were stained with antibodies specific for CD4 (clone RM4–5; eBioscience) and CD8 (clone 53–6.7; BD). Prior to running the samples, propidium iodide (Sigma-Aldrich) as live/dead marker and counting beads (Spherotech, USA) were included. Division index was defined as the average number of divisions that all cells in the initial population have undergone[73]. In some experiments, a colorimetric assay to measure the number of viable cells, CellTiter 96® AQ$_{ueous}$ One Solution, was used according to manufacturer's protocol (Promega).

**MSC culture.** Human bone marrow-derived MSCs were purchased from Tulane Center for Gene Therapy (Tulane University, New Orleans, LA). Briefly, mononuclear cells in bone marrow aspirates were separated using density gradient centrifugation, and nonadherent cells were removed from culture after 18–24 h. Adherent cells were expanded by plating at 60 cells/cm² in α-MEM (Thermo-Fisher) supplemented with 16.5% batch-tested FCS (ThermoFisher), 2 mM L-glutamine (Thermofisher), 100 U/ml penicillin and 100 mg/ml streptomycin (MSC media) in a 37 °C, 5% CO$_2$ humidified incubator. Media was changed every 3 days and cells were harvested when 70–80% confluent. Cells were detached from tissue culture flasks with TrypLE (ThermoFisher) and frozen down in α-MEM supplemented with 30% FCS and 5% DMSO. Frozen cells were cultured in MSC media for 24 h before use. Passage 3–6 cells were used in all experiments.

**Generation of BAK/BAX-deficient MSCs.** Apoptosis-deficient human MSCs were generated using Cas9 ribonucleoprotein (RNP) technology targeting BAK and BAX. The Cas9 RNP (Cas9 complexed with sgRNA targeting BAK, GGCCATGCTGGTAGACGTGT and BAX, TCTGACGGCAACTTCAACTG) (Synthego) was delivered using the P1 Primary Cell 4D-Nucleofector™ X Kit (V4XP-1032, Lonza Bioscience) and 4D-Nucleofector™ X Unit (Lonza Bioscience) following Amaxa™ 4D-Nucleofector™ protocol for undifferentiated human mesenchymal stem cells (Lonza Bioscience). Control MSCs were electroporated without sgRNA. Briefly, equal amount of sgRNA targeting BAK and BAX (1.2 pmol) were complexed with 104 pmol Cas9 enzyme in a 5 μL volume. 100,000 MSCs were resuspended in 15 μL Nucleofector™ and Supplement solution before adding to the Cas9 RNP complex to a total volume of 20uL in Nucleocuvette™. "FF104" programme in 4D-Nucleofector™ X Unit was used. After completion, prewarmed MSC medium was added to the electroporated cells before incubating for 10 mins in 37 °C, 5% CO$_2$ humidified incubator. After recovery, the cells were

cultured and expanded in the same conditions as nontargeted MSCs. Apoptosis-resistant MSCs were selectively expanded following multiple rounds of treatment with BH3-mimetic drugs (1.25 μM x 2 rounds, then 10 μM x 2 rounds), as shown in Fig. 3A.

**Induction of apoptosis.** MSCs were treated with 1.25 μM of ABT199 (BCL-2 inhibitor, iBCL2), A1331852 (BCL-XL inhibitor, iBCLxL) and S63845 (MCL-1 inhibitor, iMCL1) in MSC media for 2 h, 24 h or 72 h at 37 °C, 5% CO$_2$. Both adherent and detached cells were collected, washed and stained with AnnexinV and PI to detect apoptosis by flow cytometry. In some experiments, as indicated, MSCs were treated with 0.5 μM staurosporine (STS) (Sigma-Aldrich) in MSC medium for 6 h. MEFs were treated with 10 μM of iBCL2, iBCLxL and iMCL1 for up to 20 h.

**Detection of efferocytosis.** Human MSCs were labelled with 5uM of CellTrace™ Violet (CTV) or CellTracker™ Orange CMTMR (ThermoFisher), and fluorochrome-conjugated antihuman CD73 (clone AD2; BD Biosciences), according to manufacturer's protocol. In some experiments, human MSCs were stained with 20 ng/mL pHrodo™ RED (ThermoFisher) for 30 mins at room temperature in the dark. The labelled MSCs were then i.v. administered into BALB/c mice before the lungs harvested for analysis at various timepoints.

**Live-cell imaging.** AMs were isolated from pooled digested mouse lungs on a BD Influx™ Cell Sorter (BD Biosciences) and plated at $1 \times 10^4$ cells/well in a 96-well plate overnight. AMs were then stained with 10 mM CellTracker™ Green CMFDA (CTG) (ThermoFisher) for 30 mins at 37 °C, 5% CO$_2$. Human MSCs and BKX-MSCs were treated with BH3-mimetic drugs (1.25 μM of ABT199, A1331852 and S63845 for 2 h at 37 °C, 5% CO$_2$), labelled with 20 ng/mL pHrodo™ RED (ThermoFisher) for 30 mins at room temperature in the dark, then plated at $0.2 \times 10^5$ cells/well with the sorted AMs. To stain for dead cells, 10 mM DRAQ7 dye (Abcam) was added in the culture. Phagocytosis of apoptotic human MSCs by sorted AMs was visualised on a Leica AF6000LX (Leica Microsystems) for 24 h, imaged every 15 mins with a 10x objective. The data were analysed with ImageJ (imagej.nih.gov, 64-bit Java 1.8.0_172).

**Bioluminescence imaging.** Mice received $1 \times 10^6$ MSCs expressing firefly luciferase and green fluorescent protein[24] via tail vein injection. Ten minutes prior to imaging, mice were i.p. injected with 200 μL D-luciferin (15 mg/ml in PBS, VivoGlo Luciferin; Promega, USA. Mice were imaged under anaesthesia with 2.5% isoflurane in oxygen, or euthanised for open-chest and direct lung imaging. Imaging was performed on the Xenogen IVIS 200 or Spectrum (Perkin Elmer, USA) and analysed using the Living Image 3.2 Software (Perkin Elmer, USA). For AM depletion, mice received 60 μl of clodronate-encapsulated liposomes (5 mg/ml clodronate in liposomal structures containing 18.8 mg/ml phosphatidylcholine and 4.2 mg/ml cholesterol in PBS; Encapsula NanoSciences) via intranasal administration under light anaesthesia on days −4 and −2 prior to MSC injection on day 0.

**Immunoblotting.** MSCs were resuspended based on cell number in reducing SDS-PAGE sample buffer. Lysates were electrophoresed on Tris-glycine SDS-PAGE gels (Biorad) prior to transfer to PVDF membrane and immunoblotting for BAX (DCS Huang, WEHI), BAK (aa23–32, Sigma) or GAPDH (#2118, Cell Signalling Technologies). Following membrane washing with TBS-T, antibodies were incubated with species-specific secondary antibodies (Southern Biotech) and developed with

ECL solution (GE Lifesciences, MA). The data were recorded with Image Lab (Bio-Rad v6.1).

**Histological analysis of lung and CNS tissues**. Lung histology was performed as follows:[24] the aorta in the lower abdominal cavity was severed, and the lungs were perfused with PBS by injection into the right atrium of the heart, inflated with 10% neutral buffered formalin via the trachea, tied off, and fixed in formalin overnight. Lungs were then washed in 70% ethanol, embedded in paraffin, sectioned, and stained with H&E for detection of inflammatory cells, and periodic acid–Schiff (PAS) for detection of mucin in goblet cells (mucus-secreting cells). Histological analysis of CNS tissue was performed as follows:[27] the brain and spinal cord were dissected from mice and fixed in 10% formalin (Sigma). Serial sections (5 μm) were cut from paraffin-embedded tissues and stained with H&E to assess inflammation, and Luxol Fast Blue (LFB) for demyelination. Images were recorded and analysed with Aperio ImageScope (Leica Biosystems v12.4.3).

**RNA sequencing and bioinformatic analyses**. BALF samples were pooled from 5 mice per group for cell sorting on a BD Influx cell sorter (BD Biosciences). AMs were sorted into lysis buffer (ThermoFisher Scientific/Ambion), snap frozen in liquid nitrogen and stored at −80 °C. Each sort was performed in duplicate on separate days. RNA was isolated from sorted AMs using the Ambion RNAqueous-Micro Kit (ThermoFisher Scientific/Ambion) as per manufacturer's instructions and assessed quantitatively and qualitatively on a spectrophotometer (Nanodrop; ThermoFisher Scientific). RNA QC was performed on an Agilent Bioanalyzer 2100 with RNA Pico microfluidics chips. A low mapping rate was achieved across all samples when mapping against the human reference, ruling out human MSC contamination. Library construction was carried out using the Illumina TruSeq mRNA Library Construction Kit. Sequencing was carried out using the Illumina NextSeq500 instrument in mid-output mode. The samples for each project were multiplexed into a single lane, yielding more than 20 million reads per sample. Data were processed and normalised by RNAsik (Monash University), and loaded onto Degust (Monash University) for analysis. A False Discovery Rate (FDR) of 0.05 and a 1.5-fold change in expression value was set as a threshold of significance for differential expressed genes (DEGs). PCA plots and volcano plots were generated using Degust. Pathway association analysis was performed using the DAVID Functional Annotation tool v6.8[74]. Gene set enrichment analysis (GSEA) was performed using software v4.1.0 from the Broad Institute[75] against hallmark gene sets ($n = 50$) and C2 curated gene sets ($n = 4063$) from the Molecular Signatures Database (MSigDB). Significant enrichment was defined as a $p \leq 0.05$ and FDR < 0.25. Interferon regulated genes were identified using the Interferome v2.01 database[76]. The data are available at NCBI GEO: GSE156240.

**Statistical analysis**. All statistical analyses were conducted using GraphPad Prism v7 with alpha set to 0.05. The unpaired student's *t* test was used for comparison between two groups, and one-way independent measure ANOVA followed by Tukey's post hoc test was used for comparison between three or more groups. Lung function data and total flux data from bioluminescent images were analysed using two-way independent measure ANOVA and Tukey's post hoc test. EAE clinical scores were analysed using Kruskal-Wallis and Dunn's post hoc test. Data were represented as mean ± SEM unless otherwise stated. A *p* value of ≤ 0.05 was considered significant.

**Reporting Summary**. Further information on research design is available in the Nature Research Reporting Summary linked to this article.

# Data availability

The data supporting the findings from this study are available within the manuscript and its Supplementary Information. The RNA-sequencing data are available at GEO: GSE156240. Source data are provided with this paper. Any remaining raw data will be available from the corresponding author upon reasonable request. Source data are provided with this paper.

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

## Acknowledgements

We thank Guizhi Sun, David Hume, Ben Kile, Marco Herold, Seong Chow and Thomas Naderer for technical expertise, reagents and helpful discussions. We also acknowledge the following Platform facilities for the provision of instrumentation and technical support: Monash Animal Research Platform, FlowCore, Monash Micro Imaging, Monash Histology Platform, Micromon and Monash Bioinformatics Platform. Funding: TSPH is supported by an R.D. Wright Career Development Fellowship (APP1107188) and a Project grant (APP1162499) from the National Health and Medical Research Council of Australia. The Australian Regenerative Medicine Institute is supported by grants from the State Government of Victoria and the Australian Government.

## Author contributions

T.S.P.H. conceived, designed and performed experiments, wrote the manuscript, and secured funding. S.H.M.P. designed and performed experiments, and analysed data. J.D.R., S.M., A.H., G.W., T.B., and D.Z. performed experiments and analysed data. J.R. performed experiments. A.B. and D.P. designed experiment and analysed data. N.P. and G.D. performed experiments, analysed data and provided reagents. N.D.H., D.H. and D.H.D.G. provided reagents, expertise and feedback. All authors contributed to manuscript revision, read and approved the submitted version.

## Competing interests

T.S.P.H. received funding from Mesoblast Ltd and Regeneus Ltd outside of this work. The funders were not involved in the study design, collection, analysis, interpretation of data, the writing of this article or the decision to submit it for publication. N.D.H. and J.R. are founders and shareholders of oNKo-Innate Pty. Ltd, a discovery stage biotechnology company focussed on immuno-oncology, not related to this work. The remaining authors declare no competing interests.
