## [Peer Review File · Nature Communications]

REVIEWER COMMENTS

Reviewer #1 (Remarks to the Author):

This is a timely and well-presented investigation of whether protective effects of MSCs in a mouse model of Th2-mediated allergic airways inflammation are due to MSC apoptosis and the host immune response rather than any specific actions of the MSCs themselves. The data observed are in keeping with recent comparable literature and further make a strong case for innate immune responses mediated by macrophages as the primary effector of protective MSC actions. The manuscript is well written and the data comprehensive, compelling, and logically presented. However, there are several points for the authors to consider.

1. Allergic airways inflammation in mice has a number of pathophysiologic alterations including but not limited to increases in total BALF cell counts, increased BALF eosinophils, increased BALF Th2 cytokines, increased AHR, and peribronchial inflammatory cell infiltrates on histologic examination of the lungs. As such, it is important to assess each of these in parallel in each respective experiment, particularly as they may not always coincide. However, in the different AHR studies presented, some only had one or two of these measures and in no case was histologic data presented. Presumably all of these measures were assessed in each respective study: can the authors comment on why the full range of data wasn't included for each.

2. In parallel with growing appreciation of the role of apoptotic MSCs triggering host immune responses, there is also growing appreciation that different (lung) inflammatory environments can influence different MSC behaviors even in the absence of inducing apoptosis. As such, there are not necessarily similar immunosuppressive effects occurring in different disease states (p.4, line 58). Can the authors conceptually integrate their findings with this.

3. Further, how do the authors speculate that apoptotic MSCs work in an asthma environment as compared to for example an acute lung injury environment. For example, if the alveolar macrophages are primed for M2 behaviors in asthma and apoptotic MSCs decrease M2, and also apparently M1-related gene expression, how would this work in other inflammatory environments with M1-primed macrophages? Are there clues in refs 16-18 and 23? Is it of note that both M2 and M1 marker expression is decreased rather than a yin-yang increase in M1 markers?

4. How do these observations coincide with recent descriptions of the instant blood mediated inflammatory responses (IBMIR) following systemic administration of (allogeneic) MSCs? Did the authors have opportunity to assess changes in complement and tissue factor expression. This might

be retrievable data from the gene array studies. In parallel, was there any change in HLA ABC or DR expression by the MSCs themselves that might provoke immune recognition and clearance?

5. Appreciating that MSC apoptosis and efferocytosis occurs, is there also any change in their secretome that might further effect either their disease ameliorating effects or directly affect macrophage behavior? Arguably, this would occur within the first hour of lodging in the lungs prior to any significant apoptosis. This includes EV release which gets towards a larger concept: MSC conditioned media, in particular EVs, can be as effective as the cells themselves in ameliorating a range of disease models, including allergic airways inflammation in mice. How do the authors speculate this might fit into their model?

6. How do the authors speculate the labeled MSCs are cleared from the lungs after 72 hrs in the clodronate-treated mice? If uptake by neutrophils occurs early, something else must be occurring.

7. p. 14, line 264: “fluoresces” not “fluorescents”

8. The observation that IFN response gene expression is increased. MSC protective effects were observed to be abrogated in ova-mediated allergic airways inflammation in IFN γ receptor KO mice (Goodwin et al Stem Cells. 29(7):1137-48, 2011). How do the authors results correlate with this?

9. There is a recent review that may be appropriate for consideration that in part addresses some of the above issues: Weiss DJ, English K, Krasnodembskaya A, Isaza-Correa JM, Hawthorne IJ, Mahon BP. The Necrobiology of Mesenchymal Stromal Cells Affects Therapeutic Efficacy. *Front Immunol.* 2019 Jun 4;10:1228. doi: 10.3389/fimmu.2019.01228.

Reviewer #2 (Remarks to the Author):

Pang and colleagues (NCOMMS-20-43218) report on a biochemical, cell biological, and in vivo study that investigates the therapeutic potential and mechanisms by which applied mesenchymal stromal cells protect in a lung inflammation model of OVA-induced asthma. Investigators obtain MSC populations from bone marrow (as well as from human umbilical cord and adipose tissue as proof of concept), label cells with fluorescence or luciferase, and show that cells populate the lung and

rapidly die (mainly CD45 negative stromal cells). Authors then provide a comprehensive and rigorous study to show that caspase dependent apoptosis is required for the therapeutic benefit in the OVA-induced asthma model (as well as in an MOG inducible EAE model that recapitulates aspects of MS). Major findings also include that (i) manipulation of apoptosis by promoting apoptosis (BHS mimetics) or knocking out Bak and Bax promote or impair therapeutic effects respectively, (ii) MSC cells despite their mouse, human or allogeneic background all die and have potential therapeutic potential, and (iii) the mechanisms of protection involve alveolar macrophage efferocytosis and is independent of other lymphocytes/NK cells. Finally, authors employ in vivo efferocytosis and provide unbiased gene signatures (including IL-10). Collectively, these studies implicate dying stromal cells and subsequent efferocytosis, but not direct secretome from MSCs, as the therapeutic utility in MSC reconstitution strategies.

Overall, this is a forward-looking and important paper that identifies a new mechanism for MSC therapeutics. The data is very strong and supports the conclusions that are being by the authors. I have just two relatively comments, but all in all, this is a very nice advance to this research field.

In Fig. 1 and elsewhere, it appears a CD45 negative population is mainly responsible for the death. Can the authors provide more detail about what this cell is and why it is so sensitive to apoptosis in vivo (particularly the mitochondrial pathway of apoptosis)?

The data in Fig 5 also requires a bit more clarity. In this capacity, I am not convinced that the efferocytosis is directed by resident alveolar macrophages, as opposed to a population of CD45 positive MSC myeloid cells from the bone marrow. To show this definitively, authors would have to inject a purified population of CD45 negative cells at the beginning of the treatment group.

In Fig 5, a bit more clarification on the significance of the MHC up-regulation. If cells are making IL-10 and mon-inflammatory cytokines, why are the cells becoming activated. An explanation in the discussion would suffice.

Reviewer #3 (Remarks to the Author):

The manuscript by Pang et al. entitled “Mesenchymal stromal cell apoptosis is required for their therapeutic function” describes data that suggests that the primary therapeutic mechanisms of MSCs require them to undergo apoptosis in vivo, which permits them to mediate effects via

induction of metabolic changes within macrophages, which act upon the apoptotic bodies in an effort to clear them. As presented the data could represent a paradigm shift in our understanding of the therapeutic mechanisms of MSCs in vivo. Though the data is limited to a single route of administration, namely intravenous administration of MSCs. The authors should present their findings and discussion thereof in the context of the lung only.

Reviewer comments:

1. While the authors do present data that supports the concept that the therapeutic effects of MSCs do appear to require apoptosis in the lung, it would be beneficial to demonstrate another cell type, not an MSC, fails to mediate therapeutic benefit in vivo.

2. All of the studies presented in the manuscript focus on intravenous delivery of MSCs, it would be beneficial to know whether apoptosis is central to therapeutic mechanism via alternative routes of delivery that may avoid the lung. MSCs have been shown to be therapeutic via different routes of administration in vivo.

3. For the CTV-labelled MSC studies, the authors should stipulate the levels of sensitivity of the assay, in terms of the minimal numbers of cells that can be detected. Also, they stipulate that some of the CTV is found in a CD45+ population, but fail to elaborate on what this population may represent.

4. Moreover, MSCs are notoriously good at eliminating any sort of fluorescent dye-labeled trackers, the authors should use a PCR-based approach to demonstrate that MSCs are not surviving in vivo.

5. The authors should provide more detailed information on the generation of the apoptosis-resistant MSCs.

6. MSCs have been demonstrated to impact the frequency and function of multiple lineages of immune cells, beyond just macrophages. The authors should fully discuss how other immune cell function could be altered by the proposed apoptosis/macrophage axis.

7. In regard to the EAE studies, they are incomplete and only cursorily presented in the manuscript. They need to be more fully discussed or removed from the manuscript as without a more complete analysis and presentation they fail to add a significant amount to the manuscript.

8. The authors need to explain the delay in disease onset associated with the infusion of the BKX-MSCs. The onset is delayed in this group of animals by 5-6 days, such that the kinetics do not coordinate in any way with the onset of disease in the groups infused with the other cell types. This suggests that BKX-MSc biology is changed from the inherent MSC biology or they are functionally altered beyond just BKX inhibition.

9. Without a detailed analysis of immune infiltration, lesion number, lesion size, myelin retention, etc. It is impossible to make an assessment of the therapeutic efficacy of the cells.

10. Moreover, the infusion of MSCs prior to the onset of disease signs in vivo is not a very stringent assessment of the therapeutic potential. Cells should be administered after the onset of disease, day 15 or after.

11. It would be helpful to the reader for the authors to discuss the biologic inducers of apoptosis in the infused MSCs more fully.

Reviewer #1:

This is a timely and well-presented investigation of whether protective effects of MSCs in a mouse model of Th2-mediated allergic airways inflammation are due to MSC apoptosis and the host immune response rather than any specific actions of the MSCs themselves. The data observed are in keeping with recent comparable literature and further make a strong case for innate immune responses mediated by macrophages as the primary effector of protective MSC actions. The manuscript is well written and the data comprehensive, compelling, and logically presented. However, there are several points for the authors to consider.

1. Allergic airways inflammation in mice has a number of pathophysiologic alterations including but not limited to increases in total BALF cell counts, increased BALF eosinophils, increased BALF Th2 cytokines, increased AHR, and peribronchial inflammatory cell infiltrates on histologic examination of the lungs. As such, it is important to assess each of these in parallel in each respective experiment, particularly as they may not always coincide. However, in the different AHR studies presented, some only had one or two of these measures and in no case was histologic data presented. Presumably all of these measures were assessed in each respective study: can the authors comment on why the full range of data wasn't included for each.

We appreciate the need to multiple measures of allergic airway inflammation. Unfortunately, COVID-related disruptions and periodic, strict lockdowns in Melbourne restricted our laboratory activities and limited the breadth of our experimental analyses in our initial submission. We have since generated the remaining data, including histological analyses. These data are now presented in **revised Fig. 2(B, G), 3(E, G), 7(F, G) and new Fig. S2(B-E)**, and added to the text accordingly. These new data support our previous findings and consolidate our conclusions.

2. In parallel with growing appreciation of the role of apoptotic MSCs triggering host immune responses, there is also growing appreciation that different (lung) inflammatory environments can influence different MSC behaviors even in the absence of inducing apoptosis. As such, there are not necessarily similar immunosuppressive effects occurring in different disease states (p.4, line 58). Can the authors conceptually integrate their findings with this.

We agree that this is an important distinction and did not mean to imply that MSCs have the same effects in different diseases. Our view is that it is unclear how MSCs have therapeutic effects in different, unrelated diseases ranging from ARDS to Crohn's Disease. To clarify this, we have amended the statement to "It remains unclear how MSCs isolated from different tissues or species could exert therapeutic effects on such a wide range of unrelated diseases" (p. 4, line 56-58).

We have also discussed how our findings fit with studies showing that different inflammatory environments differentially alter MSC function. Many of the stimuli that license MSCs can also induce apoptosis (1-3) and increase anti-inflammatory macrophages (4). Similarly, different inflammatory microenvironments can differentially alter MSC function by inducing apoptosis, increasing expression of anti-inflammatory mediators and enhancing the phagocytic capacity of macrophages (5). See Discussion (p. 26).

3. Further, how do the authors speculate that apoptotic MSCs work in an asthma environment as compared to for example an acute lung injury environment. For example, if the alveolar macrophages are primed for M2 behaviors in asthma and apoptotic MSCs decrease M2, and also apparently M1-related gene expression, how would this work in other inflammatory environments with M1-primed macrophages? Are there clues in refs 16-18 and 23? Is it of note that both M2 and M1 marker expression is decreased rather than a yin-yang increase in M1 markers?

Macrophage activation occurs on a continuum, and M1/M2 marker expression is plastic across different disease stages (6). In asthma, the traditional classification of M1 macrophages as pro-inflammatory and M2 macrophages as anti-inflammatory does not seem to apply. This is because M2 genes are highly expressed in asthma, due to the asthmatic lung environment being rich in IL-4/IL-13 (7). Instead of being anti-inflammatory, the M2 genes are involved in driving asthma and lung inflammation. Rather than skewing M1/M2 expression per se, MSC treatment decreases the expression of genes known to drive asthma and lung inflammation. These genes are mostly M2, but some are M1 (Fig. 6D). Thus, we speculate that apoptotic MSCs would be similarly anti-inflammatory in other inflammatory environments with M1-primed

macrophages, as shown in models of acute lung injury, sepsis and GvHD (4, 8-11). We have now included this point in the Discussion (p. 25).

4. How do these observations coincide with recent descriptions of the instant blood mediated inflammatory responses (IBMIR) following systemic administration of (allogeneic) MSCs? Did the authors have opportunity to assess changes in complement and tissue factor expression. This might be retrievable data from the gene array studies. In parallel, was there any change in HLA ABC or DR expression by the MSCs themselves that might provoke immune recognition and clearance?

We appreciate this point and have performed additional experiments and included the new data in Supplementary Materials to address it. There was no change in HLA-ABC or HLA-DR expression in BKX-MSCs (**revised Fig. S3A**). Furthermore, treatment of control MSCs and BKX-MSCs with mouse serum did not result in cell death at 10 mins or 1 h (**new data, Fig. S3E**). Cell death was only observed after 24 h of culture with serum, and to the same extent in both types of MSCs. The killing at 24 h was inhibited by heat-inactivation of serum complement. Our data indicate that complement was not the main mediator of MSC apoptosis that occurs in the lung within 1 h post-i.v. injection (**p. 10-11**). These new data accord with our finding that MSCs are similarly cleared in NOD/SCID/*Il2r γ* ^{-/-} mice which lack haemolytic complement (**Fig. 4D**). We have integrated these points in the Discussion (**p. 23-24**).

5. Appreciating that MSC apoptosis and efferocytosis occurs, is there also any change in their secretome that might further effect either their disease ameliorating effects or directly affect macrophage behavior? Arguably, this would occur within the first hour of lodging in the lungs prior to any significant apoptosis. This includes EV release which gets towards a larger concept: MSC conditioned media, in particular EVs, can be as effective as the cells themselves in ameliorating a range of disease models, including allergic airways inflammation in mice. How do the authors speculate this might fit into their model?

The MSC secretome offers an attractive cell-free therapeutic alternative and we do not dispute the effectiveness of EVs isolated *in vitro*. As the Reviewer notes, any significant contribution

in vivo would be within the first hour post-infusion before MSCs undergo apoptosis. On the other hand, apoptotic cells also release extracellular vesicles, including apoptotic bodies, which are key regulators of immunity. Recent studies have shown that extracellular vesicles isolated from apoptotic MSCs have therapeutic potential (12-15). It is therefore plausible that the apoptotic MSC secretome has a role in MSC therapy. We have included this point in the Discussion (p. 27).

6. How do the authors speculate the labeled MSCs are cleared from the lungs after 72 hrs in the clodronate-treated mice? If uptake by neutrophils occurs early, something else must be occurring.

Clodronate liposome treatment resulted in a partial (50-60% decrease in AM numbers; see Fig. 6C/D in (16)) and transient (17) depletion of AMs. The clearance of MSCs after 72 h was likely due to the recovery of AMs. We have clarified the nature of AM depletion in the Results (p. 15-16).

7. p. 14, line 264: “fluoresces” not “fluorescents”

We have corrected this typo.

8. The observation that IFN response gene expression is increased. MSC protective effects were observed to be abrogated in ova-mediated allergic airways inflammation in IFN γ receptor KO mice (Goodwin *et al* *Stem Cells*. 29(7):1137-48, 2011). How do the authors results correlate with this?

Goodwin *et al.* (ref 58) showed that MSC effects are dependent on the recipient’s ability to respond to IFN- γ . Our data showing upregulation of IFN-responsive genes in AMs from MSC-treated mice (**Fig. 6F**) strongly support this finding by Goodwin *et al.* and suggest that alveolar macrophages are a critical cell type that can respond to IFN- γ . The upregulation of MHC class II on AMs (**Fig. 5C**) further supports this interaction between AMs and IFN- γ , which is an

inducer of MHC class II expression on mononuclear phagocytes (18). We have integrated these points in the Discussion (p. 25-26).

9. There is a recent review that may be appropriate for consideration that in part addresses some of the above issues: Weiss DJ, English K, Krasnodembskaya A, Isaza-Correa JM, Hawthorne IJ, Mahon BP. *The Necrobiology of Mesenchymal Stromal Cells Affects Therapeutic Efficacy*. *Front Immunol*. 2019 Jun 4;10:1228. doi: 10.3389/fimmu.2019.01228.

We appreciate the Reviewer pointing out this contribution and we have referred to this review (13) and integrated some of the points raised in the Discussion (p. 23, 27).

Reviewer #2:

Pang and colleagues (NCOMMS-20-43218) report on a biochemical, cell biological, and in vivo study that investigates the therapeutic potential and mechanisms by which applied mesenchymal stromal cells protect in a lung inflammation model of OVA-induced asthma. Investigators obtain MSC populations from bone marrow (as well as from human umbilical cord and adipose tissue as proof of concept), label cells with fluorescence or luciferase, and show that cells populate the lung and rapidly die (mainly CD45 negative stromal cells). Authors then provide a comprehensive and rigorous study to show that caspase dependent apoptosis is required for the therapeutic benefit in the OVA-induced asthma model (as well as in an MOG inducible EAE model that recapitulates aspects of MS). Major findings also include that (i) manipulation of apoptosis by promoting apoptosis (BHS mimetics) or knocking out Bak and Bax promote or impair therapeutic effects respectively, (ii) MSC cells despite their mouse, human or allogeneic background all die and have potential therapeutic potential, and (iii) the mechanisms of protection involve alveolar macrophage efferocytosis and is independent of other lymphocytes/NK cells. Finally, authors employ in vivo efferocytosis and provide unbiased gene signatures (including IL-10). Collectively, these studies implicate dying stromal cells and subsequent efferocytosis, but not direct secretome from MSCs, as the therapeutic utility in MSC reconstitution strategies.

Overall, this is a forward-looking and important paper that identifies a new mechanism for MSC therapeutics. The data is very strong and supports the conclusions that are being by the authors. I have just two relatively comments, but all in all, this is a very nice advance to this research field.

1. In Fig. 1 and elsewhere, it appears a CD45 negative population is mainly responsible for the death. Can the authors provide more detail about what this cell is and why it is so sensitive to apoptosis in vivo (particularly the mitochondrial pathway of apoptosis)?

We appreciate the Reviewer's query and can clarify that the CD45⁻CTV^{hi} population in the lung that showed caspase 3 activation was actually the MSCs that were injected. We have included additional FACS plots to show that the injected CD45⁻CTV^{hi} cells were CD73⁺ MSCs (**revised Fig. 1A**), while the CD45⁻CTV^{lo} population contained lung epithelial cells that had engulfed CTV-labelled MSCs (**new data, Fig. S1B**). The endogenous CD45⁻CTV⁻ population did not contain MSCs and did not show caspase 3 activation (**Fig. 1B**). We have clarified this accordingly in the Results (**p. 7**).

2. The data in Fig 5 also requires a bit more clarity. In this capacity, I am not convinced that the efferocytosis is directed by resident alveolar macrophages, as opposed to a population of CD45 positive MSC myeloid cells from the bone marrow. To show this definitively, authors would have to inject a purified population of CD45 negative cells at the beginning of the treatment group.

We agree that the predominant cell type phagocytosing MSCs were Ly6G⁺ neutrophils, and Ly6C^{hi} and Ly6C^{lo} monocytes (**Fig. 5B, D**), supporting de Witte *et al.* (19). Although alveolar macrophages, interstitial macrophages and dendritic cells were also capable of phagocytosing MSCs, these were not the predominant cell types phagocytosing MSCs in the lung. However, on a functional level, the immunosuppressive effects of MSCs in the context of lung inflammation were primarily mediated by alveolar macrophages in this model. When FACSsorted from MSC-treated mice, alveolar macrophages directly suppressed OVA-specific T cell proliferation (**Fig. 7B**) and induced IL-10 production in response to OVA (**Fig. 7C**). This effect was not observed in neutrophils or monocytes FACSsorted from MSC-treated mice (**Fig. 7B, C**). On the other, monocytic cells that were co-cultured with apoptotic MSCs and had

engulfed apoptotic MSCs *in vitro*, have been shown to be immunoregulatory (19, 20). We have clarified these points in the Discussion (p. 24).

3. In Fig 5, a bit more clarification on the significance of the MHC up-regulation. If cells are making IL-10 and non-inflammatory cytokines, why are the cells becoming activated. An explanation in the discussion would suffice.

MHC II upregulation does not indicate that cells are necessarily more immunostimulatory. Rather, the upregulation of MHC class II on haematopoietic cells that had taken up CTV-labelled MSCs suggests that the phagocytes are activated, in this case to produce anti-inflammatory cytokines. MHC class II upregulation was also observed in AMs from OVA-sensitised mice that were treated with MSCs (new data, Fig. 5C right-most panel), at a time when the AMs were immunosuppressive (Fig. 7B, C). In mononuclear phagocytes, the expression of MHC class II is induced by IFN- γ (18). Thus, the upregulation of MHC class II on AMs from MSC-treated mice may reflect the interferon-responsive genes revealed in our RNA-sequencing data (Fig. 6F). We have clarified these points in the Discussion (p. 26).

Reviewer #3:

The manuscript by Pang et al. entitled “Mesenchymal stromal cell apoptosis is required for their therapeutic function” describes data that suggests that the primary therapeutic mechanisms of MSCs require them to undergo apoptosis in vivo, which permits them to mediate effects via induction of metabolic changes within macrophages, which act upon the apoptotic bodies in an effort to clear them. As presented the data could represent a paradigm shift in our understanding of the therapeutic mechanisms of MSCs in vivo. Though the data is limited to a single route of administration, namely intravenous administration of MSCs. The authors should present their findings and discussion thereof in the context of the lung only.

1. While the authors do present data that supports the concept that the therapeutic effects of MSCs do appear to require apoptosis in the lung, it would be beneficial to demonstrate another cell type, not an MSC, fails to mediate therapeutic benefit in vivo.

We have included new data showing that mouse embryonic fibroblasts (MEFs), a non-MSC fibroblastic cell, were more resistant to apoptosis induction via the mitochondrial pathway (**new data, Fig. S2A**) and failed to inhibit lung inflammation (**new data, Fig. S2B-E**). We have referred to these new data in the Results (**p. 9**) and integrated them into the Discussion (**p. 22**).

2. All of the studies presented in the manuscript focus on intravenous delivery of MSCs, it would be beneficial to know whether apoptosis is central to therapeutic mechanism via alternative routes of delivery that may avoid the lung. MSCs have been shown to be therapeutic via different routes of administration in vivo.

We appreciate this important point. In this study, we have focussed on intravenous delivery of MSCs because it is the most common route in clinical trials (21, 22). MSCs delivered via different routes have also been shown to be effective, and there is evidence that apoptosis plays a role. Galleu *et al.* (8) showed that apoptotic MSCs administered via the intraperitoneal (i.p.) route were effective in a model of GvHD. In their study, i.p. MSCs were not detected in the lung but were detected inside phagocytes in the peritoneal lymph nodes, which produced IDO to mediate the therapeutic effects. We have similarly delivered i.p. MSCs in OVA-induced asthma and found that i.p. MSCs did not survive long in the peritoneum and could not be detected elsewhere (**Fig. 1 below**), but were effective at inhibiting cardinal features of lung inflammation (**Fig. 2 below**). While preliminary, our unpublished data support Galleu *et al.* and indicate a central role for MSC apoptosis in i.v. and i.p. delivery. We feel that our data generated using the i.v. route (being the most common route in both clinical and pre-clinical studies) effectively establish that the apoptotic paradigm is an important one to consider as a mechanism for immunomodulation. Furthermore, we believe that the question of how broadly this mode-of-action applies to alternative routes and the mechanisms effecting any changes will be fertile ground for future studies.

Fig. 1: i.p. MSCs stay in the peritoneal cavity and are cleared within 2-3 days. Images representative of 5 mice per timepoint.

Fig. 2: i.p. MSCs inhibit OVA-induced allergic asthma. Mean±SEM, 4-5 mice per group. *p≤0.05, ***p≤0.001, ****p≤0.0001, one-way ANOVA, Tukey.

3. For the CTV-labelled MSC studies, the authors should stipulate the levels of sensitivity of the assay, in terms of the minimal numbers of cells that can be detected. Also, they stipulate that some of the CTV is found in a CD45⁺ population, but fail to elaborate on what this population may represent.

We thank the Reviewer for raising these points and we have added new data to show the level of detection of CTV-labelled MSCs. We were able to detect 1x10⁶ injected cells in the lung with high confidence, and as few as 1x10⁵ cells, but not less (**new data, Fig. S1A**). See Results (**p. 7**).

We found that the CTV⁺CD45⁺ population contained haematopoietic cells that are professional phagocytes, including neutrophils, monocytes and alveolar macrophages, that had taken up CTV-labelled MSCs (**Fig. 5B**). The uptake of MSCs was confirmed by pHrodo signals in these CD45⁺ phagocytic cells upon phagocytosis of pHrodo RED-labelled MSCs (**Fig. 5D, E**). There might have been a misunderstanding because we first mentioned that CTV label was detected in both CD45⁻ stromal and CD45⁺ haematopoietic populations in Fig. 1B (**p. 7**), but only presented the CD45⁺ data in Fig. 5B (**p. 15**) and in the Discussion (**p. 24**).

4. Moreover, MSCs are notoriously good at eliminating any sort of fluorescent dye-labeled trackers, the authors should use a PCR-based approach to demonstrate that MSCs are not surviving *in vivo*.

Other than fluorescent dyes, we also used a combination of techniques, including reporter proteins (16) and enzymes (luciferase; **Fig. 4A**), which are only expressed in viable, metabolically active cells (23). Importantly, the luciferase-expressing MSCs in Fig. 4 co-expresses GFP that is lost upon apoptosis (**Fig. 3 below**). Although a PCR-based approach is highly sensitive, it does not preclude detection of transcripts derived from apoptotic cells (and phagocytes can contain “passenger” transcripts that originate from engulfed apoptotic cells) (24). In studies using PCR-based approaches, analysis of human SRY (25) or Alu (26) showed a rapid loss of human DNA within the first 24 h after intravenous administration of human MSCs, according with our data using labelled MSCs. Furthermore, live MSCs could not be cultured from the lung 72 h after i.v. administration (27), confirming that MSCs were not surviving *in vivo*.

Fig. 3: Luciferase/GFP-expressing MSCs lose GFP expression upon apoptosis induction (red histogram, bottom right panel).

5. The authors should provide more detailed information on the generation of the apoptosis-resistant MSCs.

We have provided additional, detailed information on the generation of apoptosis-resistant BKX-MSCs in the Methods (p. 35).

6. MSCs have been demonstrated to impact the frequency and function of multiple lineages of immune cells, beyond just macrophages. The authors should fully discuss how other immune cell function could be altered by the proposed apoptosis/macrophage axis.

Indeed, MSCs have been shown to influence multiple immune cell types. For example, many studies (4, 28), including ours (16), have demonstrated an increase in regulatory T cells and IL-10 production in MSC-treated mice. Apoptotic cells also induce regulatory T and B cells, tolerogenic dendritic cells and immunomodulatory macrophages, which in turn produce immunosuppressive factors such as TGF- β and IL-10 (29). It is possible that the increase in Tregs and IL-10 in MSC-treated mice reflects this apoptotic MSC/macrophage axis. Another aspect of immune regulation by apoptotic cells involves their release of extracellular vesicles, including apoptotic bodies, that can modify various components of immune function (13, 30). We have integrated these points in the Discussion (p. 24).

7. In regard to the EAE studies, they are incomplete and only cursorily presented in the manuscript. They need to be more fully discussed or removed from the manuscript as without a more complete analysis and presentation they fail to add a significant amount to the manuscript.

We acknowledge this point from the Reviewer and have added new analysis of this model to strengthen the manuscript. The EAE model allows us to establish a connection between MSC apoptosis in the lung and immune responses in the peripheral lymphoid organs that subsequently impact the tissue damage that occurs in a distant organ. The lung serves as a homing niche and activation site through which myelin-reactive T cells must traffic before entering the CNS and causing damage (31). Treatments that deliver soluble antigens to the lung have also demonstrated success in EAE (32), supporting the lung as an important nexus. Therefore, we performed additional EAE experiments, as suggested by the Reviewer, to better understand how MSCs in the lung would affect EAE.

By analysing an earlier timepoint (Day 9) and a later timepoint (Day 29), we made the following observations:

(1) The ability of control MSCs and BKX-MSCs to reduce MOG-specific T-cell proliferative responses was comparable at Day 9 post-disease induction, but by Day 29 this inhibitory effect was only maintained in mice that had received control MSCs (**new data, Fig. 3J**).

(2) Control MSCs, but not BKX-MSCs, reduced circulating inflammatory Ly6C^{hi} monocytes that are required to cause tissue damage following their infiltration into the CNS (33) (**new data, Fig. 3K**).

(3) Histological analysis of spinal cord tissue at Day 29 post-disease induction showed reduced leukocyte infiltration and demyelination within the spinal cord parenchyma only in mice that received control MSCs (**new data, Fig. 3I**), supporting clinical score data (**Fig. 3H**).

Collectively, the EAE data suggest that an early innate immune response triggered by MSC apoptosis subsequently impacts the adaptive immune response that normally drives disease. Taken together, the EAE and asthma studies support our conclusion that apoptosis of MSCs is required for efficient immunosuppressive effects *in vivo*. We have added these additional data in Results (**p. 11-12** and Discussion (**p. 22**)).

8. The authors need to explain the delay in disease onset associated with the infusion of the BKX-MSCs. The onset is delayed in this group of animals by 5-6 days, such that the kinetics do not coordinate in any way with the onset of disease in the groups infused with the other cell types. This suggests that BKX-MSC biology is changed from the inherent MSC biology or they are functionally altered beyond just BKX inhibition.

The Reviewer raises an important point and we have added new data and discussion about this interesting finding. With respect to the MSC properties of the BKX-MSCs, we analysed a range of MSC features and function in BKX-MSCs and found no differences compared to their control MSCs (**Fig. S3**). Therefore, we suggest a scenario where the delay in disease onset in BKX-MSC-treated mice reflects altered interactions between the infused MSCs and host immune response, such that impeding MSC apoptosis in the lung delays (but does not prevent) the clinical manifestations of EAE. This scenario is supported by our new data showing failure of BKX-MSCs to reduce circulating inflammatory monocytes (Day 9) that infiltrate the CNS and cause damage (33), leading to subsequent failure to inhibit disease (Day 29). Precisely how

MSC apoptosis normally influences the complex EAE response will be an interesting topic for future studies. We have added these new data to the Results (p. 12) and have expanded upon our proposed mechanisms in the Discussion (p. 22).

9. Without a detailed analysis of immune infiltration, lesion number, lesion size, myelin retention, etc. It is impossible to make an assessment of the therapeutic efficacy of the cells.

We have included additional immunological and histological data, including blood immune cells, MOG-specific T cell responses, immune infiltration and demyelination (**new data, Fig. 3I-K**) to support our conclusions. See also response to #7.

10. Moreover, the infusion of MSCs prior to the onset of disease signs in vivo is not a very stringent assessment of the therapeutic potential. Cells should be administered after the onset of disease, day 15 or after.

We do not dispute the therapeutic potential of MSCs in EAE, as has been demonstrated by numerous studies, including ours (34). Rather, we had utilised EAE as a model to investigate the impact of MSC apoptosis in disease settings where the lung was not the site of inflammation. MSC administration during the priming phase of disease enabled us to establish whether there was a connection between MSC apoptosis in the lung and T cell priming in peripheral tissues. Administering MSCs later in disease, when the blood-brain barrier is broken down, would have made it difficult to determine whether the differences between MSCs versus BKX-MSCs was the result of effects in the periphery, in the CNS, or both. The mechanisms may be different depending on the timing of MSC treatment (34). Determining how the mechanisms we have uncovered here relate to later MSC-based interventions will be an interesting topic for further study, which we hope to catalyse by our publication of our novel findings.

11. It would be helpful to the reader for the authors to discuss the biologic inducers of apoptosis in the infused MSCs more fully.

We agree that this is an important point for clarity. The rapid clearance of MSCs in the absence of an adaptive immune response and cytotoxic cells (**Fig. 4C, D**), and lack of complement killing at the early time point when i.v. injected MSCs were already apoptotic in the lungs (**Fig. S3E**), suggest that MSC apoptosis in the lung is likely due to factors unrelated to immune cell function, such as nutrient/growth factor deprivation or mechanical stressors. See Results (**p. 14**) and Discussion (**p. 24**).

References

1. X. Li, B. Shang, Y. N. Li, Y. Shi, C. Shao, IFN γ and TNF α synergistically induce apoptosis of mesenchymal stem/stromal cells via the induction of nitric oxide. *Stem Cell Res Ther* **10**, 18 (2019).
2. Y. Liu, L. Wang, T. Kikuri, K. Akiyama, C. Chen, X. Xu, R. Yang, W. Chen, S. Wang, S. Shi, Mesenchymal stem cell-based tissue regeneration is governed by recipient T lymphocytes via IFN- γ and TNF- α . *Nat Med* **17**, 1594-1601 (2011).
3. B. Salaun, P. Romero, S. Lebecque, Toll-like receptors' two-edged sword: when immunity meets apoptosis. *Eur J Immunol* **37**, 3311-3318 (2007).
4. K. Nemeth, A. Leelahavanichkul, P. S. Yuen, B. Mayer, A. Parmelee, K. Doi, P. G. Robey, K. Leelahavanichkul, B. H. Koller, J. M. Brown, X. Hu, I. Jelinek, R. A. Star, E. Mezey, Bone marrow stromal cells attenuate sepsis via prostaglandin E(2)-dependent reprogramming of host macrophages to increase their interleukin-10 production. *Nat Med* **15**, 42-49 (2009).
5. S. Rolandsson Enes, A. D. Krasnodembskaya, K. English, C. C. Dos Santos, D. J. Weiss, Research Progress on Strategies that can Enhance the Therapeutic Benefits of Mesenchymal Stromal Cells in Respiratory Diseases With a Specific Focus on Acute Respiratory Distress Syndrome and Other Inflammatory Lung Diseases. *Front Pharmacol* **12**, 647652 (2021).
6. P. J. Murray, J. E. Allen, S. K. Biswas, E. A. Fisher, D. W. Gilroy, S. Goerdt, S. Gordon, J. A. Hamilton, L. B. Ivashkiv, T. Lawrence, M. Locati, A. Mantovani, F. O. Martinez, J. L. Mege, D. M. Mosser, G. Natoli, J. P. Saeij, J. L. Schultze, K. A. Shirey, A. Sica, J. Suttles, I. Udalova, J. A. van Ginderachter, S. N. Vogel, T. A. Wynn, Macrophage activation and polarization: nomenclature and experimental guidelines. *Immunity* **41**, 14-20 (2014).

7. A. P. Moreira, C. M. Hogaboam, Macrophages in allergic asthma: fine-tuning their pro- and anti-inflammatory actions for disease resolution. *J Interferon Cytokine Res* **31**, 485-491 (2011).
8. A. Galleu, Y. Riffo-Vasquez, C. Trento, C. Lomas, L. Dolcetti, T. S. Cheung, M. von Bonin, L. Barbieri, K. Halai, S. Ward, L. Weng, R. Chakraverty, G. Lombardi, F. M. Watt, K. Orchard, D. I. Marks, J. Apperley, M. Bornhauser, H. Walczak, C. Bennett, F. Dazzi, Apoptosis in mesenchymal stromal cells induces in vivo recipient-mediated immunomodulation. *Science translational medicine* **9**, (2017).
9. F. B. Liu, Q. Lin, Z. W. Liu, A study on the role of apoptotic human umbilical cord mesenchymal stem cells in bleomycin-induced acute lung injury in rat models. *Eur Rev Med Pharmacol Sci* **20**, 969-982 (2016).
10. P. H. Sung, C. L. Chang, T. H. Tsai, L. T. Chang, S. Leu, Y. L. Chen, C. C. Yang, S. Chua, K. H. Yeh, H. T. Chai, H. W. Chang, H. H. Chen, H. K. Yip, Apoptotic adipose-derived mesenchymal stem cell therapy protects against lung and kidney injury in sepsis syndrome caused by cecal ligation puncture in rats. *Stem Cell Res Ther* **4**, 155 (2013).
11. A. G. Laing, Y. Riffo-Vasquez, E. Sharif-Paghaleh, G. Lombardi, P. T. Sharpe, Immune modulation by apoptotic dental pulp stem cells in vivo. *Immunotherapy* **10**, 201-211 (2018).
12. T. K. Phan, D. C. Ozkocak, I. K. H. Poon, Unleashing the therapeutic potential of apoptotic bodies. *Biochem Soc Trans* **48**, 2079-2088 (2020).
13. D. J. Weiss, K. English, A. Krasnodembskaya, J. M. Isaza-Correa, I. J. Hawthorne, B. P. Mahon, The Necrobiology of Mesenchymal Stromal Cells Affects Therapeutic Efficacy. *Front Immunol* **10**, 1228 (2019).
14. J. Liu, X. Qiu, Y. Lv, C. Zheng, Y. Dong, G. Dou, B. Zhu, A. Liu, W. Wang, J. Zhou, S. Liu, S. Liu, B. Gao, Y. Jin, Apoptotic bodies derived from mesenchymal stem cells promote cutaneous wound healing via regulating the functions of macrophages. *Stem Cell Res Ther* **11**, 507 (2020).
15. C. Zheng, B. Sui, X. Zhang, J. Hu, J. Chen, J. Liu, D. Wu, Q. Ye, L. Xiang, X. Qiu, S. Liu, Z. Deng, J. Zhou, S. Liu, S. Shi, Y. Jin, Apoptotic vesicles restore liver macrophage homeostasis to counteract type 2 diabetes. *J Extracell Vesicles* **10**, e12109 (2021).
16. L. J. Mathias, S. M. Khong, L. Spyroglou, N. L. Payne, C. Siatskas, A. N. Thorburn, R. L. Boyd, T. S. Heng, Alveolar macrophages are critical for the inhibition of allergic asthma by mesenchymal stromal cells. *J Immunol* **191**, 5914-5924 (2013).

17. T. Thepen, N. Van Rooijen, G. Kraal, Alveolar macrophage elimination in vivo is associated with an increase in pulmonary immune response in mice. *J Exp Med* **170**, 499-509 (1989).
18. A. Celada, R. A. Maki, The expression of I-A correlates with the uptake of interferon-gamma by macrophages. *Eur J Immunol* **19**, 205-208 (1989).
19. S. F. H. de Witte, F. Luk, J. M. Sierra Parraga, M. Gargasha, A. Merino, S. S. Korevaar, A. S. Shankar, L. O'Flynn, S. J. Elliman, D. Roy, M. G. H. Betjes, P. N. Newsome, C. C. Baan, M. J. Hoogduijn, Immunomodulation By Therapeutic Mesenchymal Stromal Cells (MSC) Is Triggered Through Phagocytosis of MSC By Monocytic Cells. *Stem Cells* **36**, 602-615 (2018).
20. T. S. Cheung, A. Galleu, M. von Bonin, M. Bornhauser, F. Dazzi, Apoptotic mesenchymal stromal cells induce prostaglandin E2 in monocytes: implications for the monitoring of mesenchymal stromal cell activity. *Haematologica* **104**, e438-e441 (2019).
21. H. Caplan, S. D. Olson, A. Kumar, M. George, K. S. Prabhakara, P. Wenzel, S. Bedi, N. E. Toledano-Furman, F. Triolo, J. Kamhieh-Milz, G. Moll, C. S. Cox, Jr., Mesenchymal Stromal Cell Therapeutic Delivery: Translational Challenges to Clinical Application. *Front Immunol* **10**, 1645 (2019).
22. J. Giri, J. Galipeau, Mesenchymal stromal cell therapeutic potency is dependent upon viability, route of delivery, and immune match. *Blood Adv* **4**, 1987-1997 (2020).
23. C. H. Contag, M. H. Bachmann, Advances in in vivo bioluminescence imaging of gene expression. *Annu Rev Biomed Eng* **4**, 235-260 (2002).
24. C. Lantz, B. Radmanesh, E. Liu, E. B. Thorp, J. Lin, Single-cell RNA sequencing uncovers heterogeneous transcriptional signatures in macrophages during efferocytosis. *Sci Rep* **10**, 14333 (2020).
25. J. Leibacher, K. Dauber, S. Ehser, V. Brixner, K. Kollar, A. Vogel, G. Spohn, R. Schafer, E. Seifried, R. Henschler, Human mesenchymal stromal cells undergo apoptosis and fragmentation after intravenous application in immune-competent mice. *Cytotherapy* **19**, 61-74 (2017).
26. R. H. Lee, A. A. Pulin, M. J. Seo, D. J. Kota, J. Ylostalo, B. L. Larson, L. Semprun-Prieto, P. Delafontaine, D. J. Prockop, Intravenous hMSCs improve myocardial infarction in mice because cells embolized in lung are activated to secrete the anti-inflammatory protein TSG-6. *Cell Stem Cell* **5**, 54-63 (2009).

27. E. Eggenhofer, V. Benseler, A. Kroemer, F. C. Popp, E. K. Geissler, H. J. Schlitt, C. C. Baan, M. H. Dahlke, M. J. Hoogduijn, Mesenchymal stem cells are short-lived and do not migrate beyond the lungs after intravenous infusion. *Front Immunol* **3**, 297 (2012).
28. H. Kavanagh, B. P. Mahon, Allogeneic mesenchymal stem cells prevent allergic airway inflammation by inducing murine regulatory T cells. *Allergy* **66**, 523-531 (2011).
29. P. Saas, E. Daguindau, S. Perruche, Concise Review: Apoptotic Cell-Based Therapies-Rationale, Preclinical Results and Future Clinical Developments. *Stem Cells* **34**, 1464-1473 (2016).
30. S. Caruso, I. K. H. Poon, Apoptotic Cell-Derived Extracellular Vesicles: More Than Just Debris. *Front Immunol* **9**, 1486 (2018).
31. F. Odoardi, C. Sie, K. Streyl, V. K. Ulaganathan, C. Schlager, D. Lodygin, K. Heckelsmiller, W. Nietfeld, J. Ellwart, W. E. Klinkert, C. Lottaz, M. Nosov, V. Brinkmann, R. Spang, H. Lehrach, M. Vingron, H. Wekerle, C. Flugel-Koch, A. Flugel, T cells become licensed in the lung to enter the central nervous system. *Nature* **488**, 675-679 (2012).
32. E. Saito, S. J. Gurczynski, K. R. Kramer, C. A. Wilke, S. D. Miller, B. B. Moore, L. D. Shea, Modulating lung immune cells by pulmonary delivery of antigen-specific nanoparticles to treat autoimmune disease. *Sci Adv* **6**, (2020).
33. A. L. Croxford, M. Lanzinger, F. J. Hartmann, B. Schreiner, F. Mair, P. Pelczar, B. E. Clausen, S. Jung, M. Greter, B. Becher, The Cytokine GM-CSF Drives the Inflammatory Signature of CCR2⁺ Monocytes and Licenses Autoimmunity. *Immunity* **43**, 502-514 (2015).
34. N. L. Payne, G. Sun, C. McDonald, D. Layton, L. Moussa, A. Emerson-Webber, N. Veron, C. Siatskas, D. Herszfeld, J. Price, C. C. Bernard, Distinct immunomodulatory and migratory mechanisms underpin the therapeutic potential of human mesenchymal stem cells in autoimmune demyelination. *Cell transplantation* **22**, 1409-1425 (2013).

REVIEWERS' COMMENTS

Reviewer #1 (Remarks to the Author):

The investigators have responded well to the reviewer comments and the manuscript is accordingly and significantly improved

Reviewer #3 (Remarks to the Author):

The authors have done a good job addressing the concerns raised in the previous review. The revised manuscript warrants being published, in my opinion.

Reviewer #1:

The investigators have responded well to the reviewer comments and the manuscript is accordingly and significantly improved.

Reviewer #3:

The authors have done a good job addressing the concerns raised in the previous review. The revised manuscript warrants being published, in my opinion.

We thank the reviewers for their time and positive comments. No further revisions were requested.